# TRAINING-FREE DISTRIBUTION ADAPTATION FOR DIFFUSION MODELS VIA MAXIMUM MEAN DISCREPANCY GUIDANCE

## ABSTRACT

Pre-trained diffusion models have emerged as powerful generative priors for both unconditional and conditional sample generation, yet their outputs often deviate from the characteristics of user-specific target data. Such mismatches are especially problematic in domain adaptation tasks, where only a few reference examples are available and retraining the diffusion model is infeasible. Existing inference-time guidance methods can adjust sampling trajectories, but they typically optimize surrogate objectives such as classifier likelihoods rather than directly aligning with the target distribution. We propose *MMD Guidance*, a training-free mechanism that augments the reverse diffusion process with gradients of the *Maximum Mean Discrepancy (MMD)* between generated samples and a reference dataset. MMD provides reliable distributional estimates from limited data, exhibits low variance in practice, and is efficiently differentiable, which makes it particularly well-suited for the guidance task. Our framework naturally extends to prompt-aware adaptation in conditional generation models via product kernels. Also, it can be applied with computational efficiency in latent diffusion models (LDMs), since guidance is applied in the latent space of the LDM. Experiments on synthetic and real-world benchmarks demonstrate that MMD Guidance can achieve distributional alignment while preserving sample fidelity.

## 1 INTRODUCTION

The rapid advancement of generative artificial intelligence is largely driven by the paradigm of foundation models: pre-trained neural networks capable of generalization across diverse tasks and modalities (Bommasani et al., 2021). These models, including large language models (LLMs) (Brown et al., 2020) and denoising diffusion models (Ho et al., 2020; Song et al., 2021; Rombach et al., 2022), offer powerful priors that can be adapted to specific downstream applications with minimal computational overhead. While this adaptation can occur through fine-tuning or prompt engineering, training-free inference-time methods remain relatively underexplored, despite their practical advantages of no memory overhead and immediate deployment.

A fundamental challenge in deploying generative models is *distribution matching*, i.e., aligning model outputs with a user's target distribution that differs from the pre-training corpus. For example, consider a medical imaging scenario where a practitioner needs synthetic X-rays matching their hospital's specific equipment characteristics, or a designer requiring images that follow their brand's specific visual style, where both tasks are specified through a small number (e.g. 50-100) of reference examples. A generic text-conditioned image generation model will generate samples following its learned priors, leading to a distribution mismatch. Existing methods cannot fully address the distribution mismatch without computationally expensive retraining on a sufficiently large sample set from the target distribution, which is often unavailable to typical users of generative AI services.

The mentioned gap between model capabilities and deployment requirements motivates the following question: Can we guide pre-trained diffusion models to match a user's target distribution at inference time using limited reference samples from their distribution? Note that this distribution matching task can be considered in both the prompt-free (unconditional) and prompt-aware (conditional) data generation, where the distribution alignment targets the user's distribution of interest.

**Overview of MMD Guidance Method in Iteration *n*:**

Figure 1: Image generation via a text-conditioned latent diffusion model (LDM) with no guidance vs. our proposed MMD guidance. The LDM (Stable Diffusion-XL) following our proposed MMD guidance over 100 reference samples of "cat" and "dog" images could exhibit the visual format of the target distribution, but the unguided LDM's output samples differ in style from the target model.

In particular, diffusion models are suited for such training-free adaptation due to their iterative denoising process. During the sampling phase of diffusion models, external signals can steer the generation trajectory through guidance in sample generation without modifying the parameters of the pre-trained neural net denoiser. Existing guidance techniques successfully bias samples toward specific objectives: classifier guidance (Dhariwal & Nichol, 2021) maximizes class probabilities, classifier-free guidance (Ho & Salimans, 2022) amplifies conditioning, and score distillation (Poole et al., 2023) optimizes external losses. However, none of these methods can explicitly match a target distribution defined by reference samples, since they optimize surrogate objectives rather than directly minimizing distributional discrepancy. The key challenge is how one can quantify and minimize distribution mismatch within the reverse diffusion dynamics.

In this work, we propose the *MMD Guidance* approach, an inference-time mechanism that augments the reverse diffusion process with gradients from the Maximum Mean Discrepancy (MMD) (Gretton et al., 2012). This approach aims to address distribution matching by incorporating $\nabla_{x_t}\text{MMD}^2$ into the sampling dynamics at each timestep. MMD is uniquely suited for this task. Unlike the family of $f$-divergences and Wasserstein distances, the MMD distance provides unbiased, low-variance estimates from limited samples without suffering from the curse of dimensionality. Moreover, the kernel-based formulation in MMD is efficiently differentiable, enabling scalable gradient computation with respect to generated samples.

Our proposed MMD guidance method can smoothly extend to *prompt-aware distribution matching* in text-conditioned image generation. Users often need outputs that satisfy both textual semantics ("a portrait of a person") and distributional constraints (matching the style of their reference portraits). We attempt to achieve this property through the product kernel function $k_{\text{multimodal}}([p, x], [p', x']) = k_{\text{text}}(p, p') \cdot k_{\text{image}}(x, x')$ that jointly measure similarity in prompt and visual spaces, enabling prompt-aware control without any retraining of the denoiser in conditional diffusion model sample generation.

For practical deployment, we implement MMD Guidance efficiently in the *latent space* of a *latent diffusion model (LDM)* (Rombach et al., 2022). By computing MMD and its gradients directly on the latent representation $z \in \mathcal{Z}$ rather than pixel space, we achieve a significant speedup while maintaining alignment quality. This enhances the efficiency and scalability of the MMD guidance method for state-of-the-art conditional diffusion generation models.

We validate MMD Guidance in our numerical experiments on synthetic distributions, standard image databases, and domain adaptation benchmarks. On synthetic Gaussian mixture models, the MMD guidance can recover the target mixture from a limited number of samples. In the case of image generation, the MMD Guidance framework reduces distribution discrepancy compared to the baseline training-free guidance methods while maintaining comparable fidelity scores in image generation. Also, the MMD guidance method could preserve sample quality through inference-time guidance in the latent space with proper generalization. Figure 1 illustrates an example of applying the prompt-aware MMD guidance with only 100 reference samples of "cat" and "dog" categories to the latent space of the SD-XL (Podell et al., 2023) latent diffusion model. In summary, our work's main contributions are:

- We introduce MMD Guidance, a training-free method for inference-time adaptation of diffusion models to a target distribution specified by a limited reference set.
- We propose a practical extension for prompt-aware adaptation using product kernels, enabling joint alignment to both text prompts and visual styles.
- We develop an efficient implementation that operates directly in the latent space of LDMs, ensuring scalability to state-of-the-art conditional diffusion models.
- We provide empirical validation demonstrating that MMD guidance can achieve satisfactory distribution matching.

## 2 RELATED WORK

**Guidance in diffusion models.** Inference-time guidance has been central to the success of diffusion models. Classifier guidance augments reverse diffusion with classifier gradients, boosting conditional fidelity at the expense of diversity (Dhariwal & Nichol, 2021). Classifier-free guidance (CFG) removes the external classifier by interpolating conditional and unconditional scores (Ho & Salimans, 2022). This line of work has been extended in several ways. Malarz et al. (2025) introduce $\beta$-CFG, which controls the impact of guidance during the denoising process. Similarly, CFG++ (Chung et al., 2025) aims to mitigate the off-manifold behavior associated with the standard CFG. Follow-ups extend guidance to multiple conditions via product-of-experts composition (Liu et al., 2022) and to editing through cross-attention control or partial noising (Hertz et al., 2023; Meng et al., 2022). These approaches bias sampling toward predefined conditions, but do not explicitly align generations to an arbitrary reference distribution.

**Conditional Generation with Guidance.** Controlling generative processes with specific conditions is increasingly vital for practical applications, utilizing inputs such as text guidance (Kim et al., 2022; Nichol et al., 2022; Liu et al., 2023), class labels (Dhariwal & Nichol, 2021), style images (Mou et al., 2024; Zhang et al., 2023) and human motions (Tevet et al., 2023). Conditional generation methods are divided into training-based and training-free approaches. Training-based methods either learn a time-dependent classifier that guides the noisy sample $xt$ toward the condition $y$ (Dhariwal & Nichol, 2021; Nichol et al., 2022; Zhao et al., 2022; Liu et al., 2023) or directly train a conditional denoising model $\epsilon_\theta(x_t, t, y)$ through few-shot adaptation (Mou et al., 2024; Rombach et al., 2022; Ruiz et al., 2023). In contrast, training-free guidance enables zero-shot conditional generation using a pre-trained differentiable predictor, such as a classifier or energy function, to assess the alignment of generated samples with target conditions (He et al., 2024; Bansal et al., 2023; Yu et al., 2023; Ye et al., 2024). Our work is a training-free approach that uses maximum mean discrepancy to improve distributional alignment while improving the sample diversity.

**Adaptation of Diffusion Models.** A complementary line adapts pretrained text-to-image models to user-specific concepts and styles. Textual Inversion learns novel token embeddings for new subjects (Gal et al., 2022); DreamBooth fine-tunes for subject fidelity with a handful of images (Ruiz et al., 2023); and LoRA enables parameter-efficient adaptation via low-rank updates (Hu et al., 2022). Structural controllers such as ControlNet and T2I-Adapter add spatial/structural conditions

with lightweight modules (Zhang & Agrawala, 2023; Mou et al., 2024). In a recent work, Domain Guidance (Zhong et al., 2025) uses the pretrained diffusion models and a fine-tuned model to guide generation toward the target domain. While effective, these methods require optimization and parameter storage, and can overfit tiny reference sets. Our approach is training-free and operates at inference time, making it complementary in deployment and privacy-constrained settings.

**MMD and distribution alignment.** The maximum mean discrepancy (MMD) is a kernel-based integral probability metric with strong finite-sample properties, widely used in two-sample testing (Gretton et al., 2012), domain adaptation (Long et al., 2015), and generative modeling (Li et al., 2015; 2017; Bińkowski et al., 2018). In concurrent work, Galashov et al. (2025) replaces the static guidance weight $w$ with a timestep-dependent learnable function. They train a neural network to predict this function using MMD as the optimization objective. In contrast, our training-free method applies MMD directly within the denoising process by injecting $\nabla_{x_t}\text{MMD}^2$ into the diffusion sampler, guiding samples toward the reference distribution, whereas prior work primarily employs MMD as a training loss or evaluation metric.

## 3 PRELIMINARIES

### 3.1 KERNEL FUNCTIONS AND MAXIMUM MEAN DISCREPANCY (MMD)

Let $k : \mathcal{X} \times \mathcal{X} \to \mathbb{R}$ be a positive semi-definite kernel with associated reproducing kernel Hilbert space (RKHS) $\mathcal{H}_k$ and feature map $\varphi : \mathcal{X} \to \mathcal{H}_k$ satisfying $k(x, x') = \langle \varphi(x), \varphi(x') \rangle_{\mathcal{H}_k}$. The *kernel mean embedding* of a distribution $P$ over $\mathcal{X}$ is defined as:

$$\mu_P := \mathbb{E}_{x \sim P}[\varphi(x)] \in \mathcal{H}_k. \tag{1}$$

The *Maximum Mean Discrepancy (MMD)* between distributions $P$ and $Q$ measures the RKHS distance between their mean embeddings (Gretton et al., 2012):

$$\text{MMD}^2(P, Q) = \left\| \mu_P - \mu_Q \right\|_{\mathcal{H}_k}^2 = \mathbb{E}_{x,x' \overset{\text{iid}}{\sim} P}\big[k(x, x')\big] + \mathbb{E}_{y,y' \overset{\text{iid}}{\sim} Q}\big[k(y, y')\big] - 2 \cdot \mathbb{E}_{x \sim P, y \sim Q}\big[k(x, y)\big].$$

For characteristic kernels (e.g., Gaussian kernel), MMD defines a metric on probability measures: $\text{MMD}(P, Q) = 0$ if and only if $P = Q$, and it always satisfies the triangle inequality.

### 3.2 LATENT DIFFUSION MODEL (LDM)

Latent Diffusion Models (LDMs) (Rombach et al., 2022) perform diffusion in the latent space of a pre-trained variational autoencoder (VAE), achieving computational efficiency while maintaining generation quality. Given encoder $\mathcal{E} : \mathcal{X} \to \mathcal{Z}$ and decoder $\mathcal{D} : \mathcal{Z} \to \mathcal{X}$, LDMs operate on latent codes $z = \mathcal{E}(x)$ in the latent space $\mathcal{Z}$. The forward diffusion process progressively adds Gaussian noise to latents:

$$q(z_t|z_0) = \mathcal{N}(z_t; \sqrt{\bar{\alpha}_t}z_0, (1 - \bar{\alpha}_t)I), \quad z_t = \sqrt{\bar{\alpha}_t}z_0 + \sqrt{1 - \bar{\alpha}_t}\epsilon_t, \tag{2}$$

where $\epsilon_t \sim \mathcal{N}(0, I)$, $\bar{\alpha}_t = \prod_{s=1}^{t} \alpha_s$, and the parameters $\{\alpha_t\}_{t=1}^{T}$ follow a proper schedule. A denoising U-Net $\epsilon_\theta(z_t, t, c)$ is trained to predict the noise $\epsilon$ given noisy latent $z_t$, timestep $t$, and optional conditioning $c$ (e.g., text embeddings from CLIP). The training objective is $\mathcal{L}(\theta) = \mathbb{E}_{z_0, \epsilon, t, c}\big[\|\epsilon - \epsilon_\theta(z_t, t, c)\|^2\big]$. In the *sampling* phase, we proceed via iterative denoising starting from $z_T \sim \mathcal{N}(0, I)$:

$$z_{t-1} = \frac{1}{\sqrt{\alpha_t}}\left(z_t - \frac{1 - \alpha_t}{\sqrt{1 - \bar{\alpha}_t}}\epsilon_\theta(z_t, t, c)\right) + \sigma_t \eta, \tag{3}$$

where $\eta \sim \mathcal{N}(0, I)$, $\sigma_t^2 = \tilde{\beta}_t = \frac{1 - \bar{\alpha}_{t-1}}{1 - \bar{\alpha}_t}\beta_t$ for DDPM sampling (Ho et al., 2020). The *guidance mechanisms* modify the predicted mean $\mu_\theta$ to steer generation. Notably, the classifier-free guidance (Ho & Salimans, 2022) interpolates conditional and unconditional predictions as follows, in which $w > 1$ amplifies conditioning strength,

$$\tilde{\epsilon}_\theta(z_t, t, c) = (1 - w) \cdot \epsilon_\theta(z_t, t, \varnothing) + w \cdot \epsilon_\theta(z_t, t, c) \tag{4}$$

## 4 MMD GUIDANCE FOR UNCONDITIONAL DIFFUSION MODELS

### 4.1 DISTRIBUTION MATCHING VIA DIVERGENCE GUIDANCE IN DIFFUSION MODELS

Consider the task of adapting a pre-trained diffusion model to generate samples matching a target distribution $Q$, specified only through a small set of reference samples $\mathcal{R} = \{x_j^{(r)}\}_{j=1}^{N_r}$. In our work, we aim to develop a guidance-based framework to address this challenge and perform a training-free adaptation of the pre-trained diffusion model. To guide the sampling process toward this target, we propose augmenting the reverse diffusion with gradients of a divergence measure between the distributions of generated and reference samples.

The choice of divergence measure is critical when estimating from limited data. Standard $f$-divergences such as the KL-divergence and total variation distance suffer from the curse of dimensionality in high-dimensional spaces, requiring sample complexity exponential in dimension for reliable estimation (Sriperumbudur et al., 2012). Similarly, Wasserstein distances, while providing a powerful metric in distinguishing probability distributions, exhibit poor sample complexity in moderately high dimensions, with known minimax estimation rates scaling as $n^{-1/d}$ for $d$-dimensional data (Weed & Bach, 2019).

We propose using Maximum Mean Discrepancy (MMD) as the divergence measure in the distribution matching guidance process. The choice of MMD distance offers two key advantages: (i) sample complexity independent of ambient dimension for characteristic kernels, and (ii) closed-form gradient computation enabling efficient optimization. More specifically, considering samples $\{z_t^{(i)}\}_{i=1}^B$ at timestep $t$ with empirical distribution $\widehat{P}_t = \frac{1}{B} \sum_{i=1}^B \delta_{z_t^{(i)}}$ and reference distribution $\widehat{Q} = \frac{1}{N_r} \sum_{j=1}^{N_r} \delta_{z_j^{(r)}}$, the empirical squared MMD is:

$$\widehat{\text{MMD}}^2(\widehat{P}_t, \widehat{Q}) = \frac{1}{B^2} \sum_{i,i'=1}^B k(z_t^{(i)}, z_t^{(i')}) + \frac{1}{N_r^2} \sum_{j,j'=1}^{N_r} k(z_j^{(r)}, z_{j'}^{(r)}) - \frac{2}{BN_r} \sum_{i=1}^B \sum_{j=1}^{N_r} k(z_t^{(i)}, z_j^{(r)}).$$

### 4.2 MMD-GUIDED SAMPLING IN DIFFUSION MODELS

We incorporate MMD minimization into the diffusion sampling process by modifying the reverse trajectory with distribution-aware gradients. The MMD-guided sampling update will thus become:

$$z_{t-1}^{(i)} = \text{sampler}(z_t^{(i)}, t, \epsilon_\theta) - \lambda_t \nabla_{z_t^{(i)}} \widehat{\text{MMD}}^2(\widehat{P}_t, \widehat{Q}), \tag{5}$$

where $\text{sampler}(\cdot)$ denotes any standard sampling scheme (e.g. DDPM or DDIM), and $\lambda_t$ controls guidance strength. Note that we subtract the MMD-squared gradient in the guidance process, as we seek to *minimize* the MMD between distributions.

**Remark 1.** *For computational efficiency, we perform MMD guidance directly in the* latent space *of latent diffusion models (LDMs). Given reference images $\{x_j^{(r)}\}_{j=1}^{N_r}$, we encode them once as $z_j^{(r)} = \mathcal{E}(x_j^{(r)})$ using the common VAE encoder of latent diffusion models (LDMs) (Rombach et al., 2022). This latent-space guidance offers two advantages: 1) computational efficiency of operating in a lower dimension of the standard latent spaces of LDMs compared to the original space, 2) statistical efficiency of the compressed latent representation capturing semantic structure while filtering pixel-level noise, improving the signal-to-noise ratio for MMD estimation. Algorithm 1 contains the main steps of the MMD-guided diffusion sample generation in the latent space of an LDM.*

For the gradient computation with respect to sample $z_t^{(i)}$, only terms in the MMD-squared containing $z_t^{(i)}$ contribute non-zero gradients. Hence, assuming a differentiable kernel $k$, this approach yields:

$$\nabla_{z_t^{(i)}} \widehat{\text{MMD}}^2(\widehat{P}_t, \widehat{Q}) = \underbrace{\frac{2}{B^2} \sum_{j=1}^B \nabla_{z_t^{(i)}} k(z_t^{(i)}, z_t^{(j)})}_{\text{intra-batch term}} - \underbrace{\frac{2}{BN_r} \sum_{j=1}^{N_r} \nabla_{z_t^{(i)}} k(z_t^{(i)}, z_j^{(r)})}_{\text{cross term with references}}, \tag{6}$$

---

**Algorithm 1:** MMD-Guided Diffusion Sampling

---

**Input:** Reference data $\mathcal{R} = \{x_j^{(r)}\}_{j=1}^{N_r}$, batch size $B$, guidance schedule $\{\lambda_t\}_{t=1}^T$, denoiser $\epsilon_\theta$

**Output:** Generated samples $\{x^{(i)}\}_{i=1}^B$ matching target distribution

*Preprocessing:* $z_j^{(r)} \leftarrow \mathcal{E}(x_j^{(r)})$ for all $j \in [N_r]$

*Initialization:* $z_T^{(i)} \sim \mathcal{N}(0, I)$ for all $i \in [B]$;

**for** $t = T$ **to** 1 **do**
    **for** $i = 1$ **to** $B$ *in parallel* **do**
        $\widehat{z}_{t-1}^{(i)} \leftarrow \text{sampler}(z_t^{(i)}, t, \epsilon_\theta)$
        $g^{(i)} \leftarrow \nabla_{z_t^{(i)}} \widehat{\text{MMD}}^2(\widehat{P}_t, \widehat{Q})$ `// Compute via equation 6`
        $z_{t-1}^{(i)} \leftarrow \widehat{z}_{t-1}^{(i)} - \lambda_t g^{(i)}$

**return** $\{x^{(i)} = \mathcal{D}(z_0^{(i)})\}_{i=1}^B$

---

where $\nabla_{z_t^{(i)}}$ denotes the gradient with respect to the samples $z_t^{(i)}$. For example, considering the Gaussian kernel $k(u,v) = \exp(\frac{-\|u-v\|_2^2}{2\sigma^2})$, we will have have $\nabla_u k(u,v) = \frac{k(u,v)}{\sigma^2}(v-u)$, yielding:

$$\nabla_{z_t^{(i)}} \widehat{\text{MMD}}^2(\widehat{P}_t, \widehat{Q}) = -\frac{2}{\sigma^2 B^2} \sum_{j=1}^B k(z_t^{(i)}, z_t^{(j)})(z_t^{(i)} - z_t^{(j)}) + \frac{2}{\sigma^2 B N_r} \sum_{j=1}^{N_r} k(z_t^{(i)}, z_j^{(r)})(z_t^{(i)} - z_j^{(r)}).$$

Under the minimization update $z \leftarrow z - \lambda_t \nabla \widehat{\text{MMD}}^2$, the intra-batch term creates *repulsion* among generated samples (promoting diversity), while the cross term creates *attraction* toward the reference samples (encouraging distribution matching). Note that we primarily need the cross term to provide a reliable gradient toward the target distribution, which we show in the following theorem.

**Theorem 1** (Concentration of Cross Term in MMD Gradient). *Consider sample space $\mathcal{Z} \subseteq \mathbb{R}^d$. Let $k : \mathcal{Z} \times \mathcal{Z} \to \mathbb{R}$ be a normalized kernel with $k(z,z) = 1$ for all $z \in \mathcal{Z}$. Suppose $k$ is differentiable and L-Lipschitz w.r.t. either input, i.e., $|k(z,w) - k(z',w)| \leq L\|z - z'\|_2$ for all $z, z', w \in \mathcal{Z}$. Let $Q$ be the target distribution on $\mathcal{Z}$ and let $\{z_j^{(r)}\}_{j=1}^{N_r} \overset{iid}{\sim} Q$ be reference samples. For every $z_0 \in \mathcal{Z}$, we define the population cross term and the empirical cross term as follows:*

$$g_{cross}^*(z_0) = -2\,\mathbb{E}_{z' \sim Q}[\nabla_{z_0} k(z_0, z')], \quad \widehat{g}_{cross}(z_0) = -\frac{2}{N_r} \sum_{j=1}^{N_r} \nabla_{z_0} k(z_0, z_j^{(r)}). \tag{7}$$

*Then for every $\delta > 0$, with probability at least $1 - \delta$ over the draw of reference samples we have:*

$$\left\|\widehat{g}_{cross}(z_0) - g_{cross}^*(z_0)\right\|_2 \leq \frac{4L}{\sqrt{N_r}}\left(1 + \sqrt{2\log(1/\delta)}\right). \tag{8}$$

**Remark 2.** *Theorem 1's bound in (8) holds with probability at least $1 - \delta$. Over $T$ iterations, ensuring the bound at each iteration to holds with probability $1 - \delta/T$ and applying a union bound implies, with probability at least $1 - \delta$, the bound holds simultaneously for all iterations $1 \leq i \leq T$:*

$$\left\|\widehat{g}_{cross}(z_i) - g_{cross}^*(z_i)\right\|_2 \leq \frac{4L}{\sqrt{N_r}}\left(1 + \sqrt{2\log(T/\delta)}\right). \tag{9}$$

**Corollary 1** (Gaussian RBF Kernel Concentration). *For the Gaussian RBF kernel $k(x,y) = \exp(-\|x-y\|_2^2/2\sigma^2)$, the cross term satisfies the following with probability at least $1 - \delta$:*

$$\left\|\widehat{g}_{cross}(z_0) - g_{cross}^*(z_0)\right\|_2 \leq \frac{3}{\sigma\sqrt{N_r}}\left(1 + \sqrt{2\log(1/\delta)}\right). \tag{10}$$

Theorem 1 provides pointwise concentration guarantees for a fixed latent point $z_0 \in \mathcal{Z}$. We extend this result to obtain uniform concentration over the norm ball in the latent space, guaranteeing that the MMD guidance gradient concentrates simultaneously for all latent vectors visited by the sampler.

**Theorem 2** (Uniform concentration of the gradients). *Consider the setting of Theorem 1 and let* $\mathcal{Z} = \{z \in \mathbb{R}^d : \|z\|_2 \leq R\}$. *Suppose* $\|\nabla_z k(z, w) - \nabla_z k(z', w)\|_2 \leq L' \|z - z'\|_2$ *for all* $z, z', w \in \mathcal{Z}$. *For every* $\delta > 0$, *the following holds with probability at least* $1 - \delta$

$$\sup_{z \in \mathcal{Z}} \left\| \widehat{g}_{\text{cross}}(z) - g^*_{\text{cross}}(z) \right\|_2 \leq \frac{4L'}{\sqrt{N_r}} + \frac{4L}{\sqrt{N_r}} \left( 1 + \sqrt{2d \log\left(6R\sqrt{N_r}\right) + 2\log\left(1/\delta\right)} \right).$$

## 5 Prompt-Aware MMD Guidance in Conditional Diffusion Models

Text-conditioned diffusion models can also be adapted to the distribution of a set of reference (prompt,data) pairs. Here, we extend MMD guidance to this setting by defining a joint divergence over the product space $\mathcal{P} \times \mathcal{Z}$ of prompts and latents. Our approach is to consider a product kernel that decomposes similarity into semantic and visual components:

$$k_{\otimes}([p, z], [p', z']) = k_p(p, p') \cdot k_z(z, z'), \tag{11}$$

where $k_p : \mathcal{P} \times \mathcal{P} \to \mathbb{R}$ measures semantic similarity between prompt embeddings and $k_z : \mathcal{Z} \times \mathcal{Z} \to \mathbb{R}$ measures visual similarity between latents. As shown in (Bamberger et al., 2022; Wu et al., 2025), this product kernel corresponds to the tensor product feature map $\varphi_{\otimes}(p, z) = \varphi_p(p) \otimes \varphi_z(z)$ in the product RKHS $\mathcal{H}_p \otimes \mathcal{H}_z$. The induced MMD in this space captures distributional differences in both modalities simultaneously.

Given generated pairs $\{(p_i, z_t^{(i)})\}_{i=1}^B$ with distribution $\widehat{P}_t$ and reference pairs $\{(p_j^{(r)}, z_j^{(r)})\}_{j=1}^{N_r}$ with distribution $\widehat{Q}$, the gradient with respect to $z_t^{(i)}$ factors through the product structure:

$$\nabla_{z_t^{(i)}} \widehat{\text{MMD}}_{\otimes}^2(\widehat{P}_t, \widehat{Q}) = \frac{2}{B^2} \sum_{j=1}^B k_p(p_i, p_j) \nabla_{z_t^{(i)}} k_z(z_t^{(i)}, z_t^{(j)}) - \frac{2}{BN_r} \sum_{j=1}^{N_r} k_p(p_i, p_j^{(r)}) \nabla_{z_t^{(i)}} k_z(z_t^{(i)}, z_j^{(r)}). \tag{12}$$

The prompt kernel $k_p(p_i, p_j^{(r)})$ acts as an attention weight: reference samples with semantically similar prompts contribute more strongly to the guidance signal. We present the steps of the resulting prompt-aware MMD guidance for prompt-conditioned diffusion models in Algorithm 2.

## 6 Numerical Results

We evaluated MMD Guidance as a training-free adaptation method in two scenarios: (1) prompt-free (unconditional) distribution alignment, (2) Prompt-aware adaptation for text-conditioned latent diffusion models. In our evaluation, we consider the comparison against these baselines: (1) unguided diffusion sampling (No-Guidance), (2) classifier-free guidance (CFG) (3) classifier guidance (CG), where we utilize a binary classifier trained to distinguish the user's reference dataset from the samples of the original diffusion model. We also report training-based baselines in Appendix C.

**Models and Settings.** We conducted the experiments on the prompt-free latent diffusion models (LDM) (Rombach et al., 2022), and Stable Diffusion v1.4 (Rombach et al., 2022). For prompt-aware experiments, we used Stable Diffusion XL (Podell et al., 2023), and PixArt (Chen et al., 2023). In all the cases, the guidance is applied in the latent space as we discussed in Remark 1.

**Evaluation.** We evaluated the generated samples based on fidelity and distributional coverage. For fidelity, we measure Fréchet distance (FD) (Heusel et al., 2017) and kernel distance (KD) (Bińkowski et al., 2018). For distributional coverage, we report Coverage/Density (Naeem et al., 2020), and RRKE (Jalali et al., 2023). All metrics are reported as mean over 5 random seeds, and we used DINOv2 (Oquab et al., 2023) as the image embedding, following the study by Stein et al. (2023).

### 6.1 MMD Guidance for Unconditional Diffusion Models

**Synthetic datasets.** We tested the MMD guidance and other baselines on synthetic data drawn from a 8-modal Gaussian Mixture Model (GMM) with eight components. Figure 2 report metrics on a simulated user with 200 data uniformly drawn from the four orange-colored components. As reported in Figure 2, the MMD Guidance method led to the best FD and KD scores. Additional experimental results on several other synthetic GMMs are presented in Appendix C.

**Changing Mode Proportions in a Mixture of Gaussians.** To examine whether MMD guidance can correct a mismatch in mixture weights between the training distribution and a target reference distribution, we use an 8-component GMM with uniformly weighted training samples. When sampling

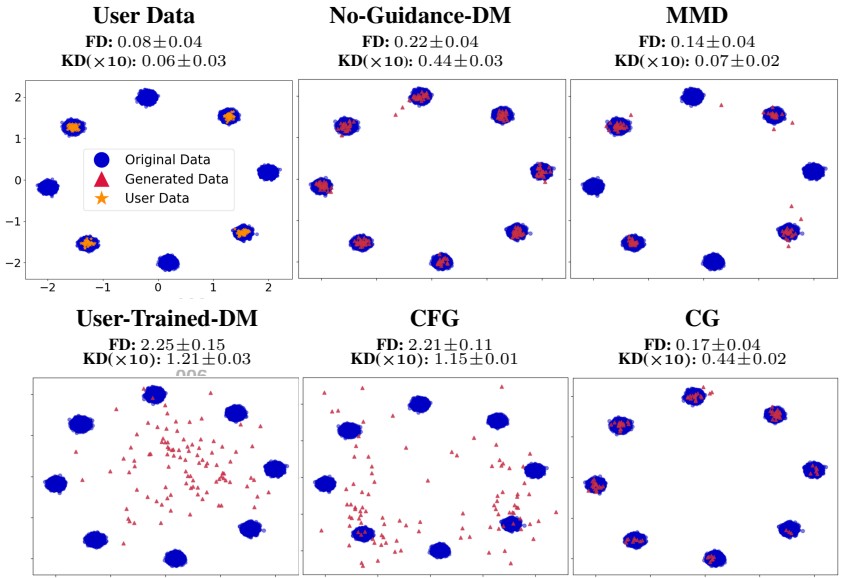

Figure 2: Comparison of MMD guidance with baselines on 100D Gaussian distributions, when guiding toward a user with 4 Gaussian components.

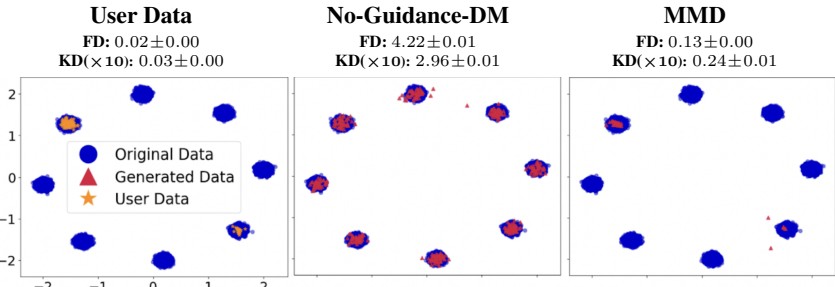

Figure 3: Effect of mode proportions in MMD guidance.

from the model without any guidance, they remain close to uniform and therefore do not match an imbalanced target distribution. To define the reference distribution, we randomly select two mixture components and sample their mixture weights from a Dirichlet distribution, yielding a highly imbalanced mixture where one component dominates the other, and then apply MMD guidance using samples from this reference distribution. As shown in Figure 3, MMD guidance not only steers samples toward the intended high-probability components but also closely reproduces the target mixture proportions of the reference distribution. Numerical evaluations are included in Appendix C

**Real-image datasets.** We also tested the MMD Guidance on real image dataset benchmarks, including FFHQ (Karras et al., 2019), and CelebA-HQ (Karras et al., 2017), using pre-trained LDMs of (Rombach et al., 2022), and a mixture-type image dataset generated with Stable Diffusion v1.4 using the prompts *car* and *bike* under four different style variations (black-and-white, winter scenes, sketch, and cartoon). We conducted experiments with two simulated users, each with 500 reference samples of the FFHQ dataset of the following specific styles: (User 1) people wearing sunglasses and (User 2) people wearing reading glasses. Table 1 reports the measured scores, where MMD Guidance attained the highest fidelity and distributional coverage scores. The randomly generated data for the qualitative evaluation are shown in Figure 4, which support the quantitative comparison. The additional results on FFHQ and other datasets as well as ablation results are in Appendix C.

## 6.2 PROMPT-AWARE MMD GUIDANCE

**Prompt-aware MMD Guidance in LDMs.** To further assess the effectiveness of MMD Guidance in conditional diffusion models, we constructed a dataset where categories were paired with distinct visual styles: cats with anime, dogs with Van Gogh paintings, cars with Pixar animation, and horses with cowboy movie styles. Prompts were generated using GPT-5, and the corresponding

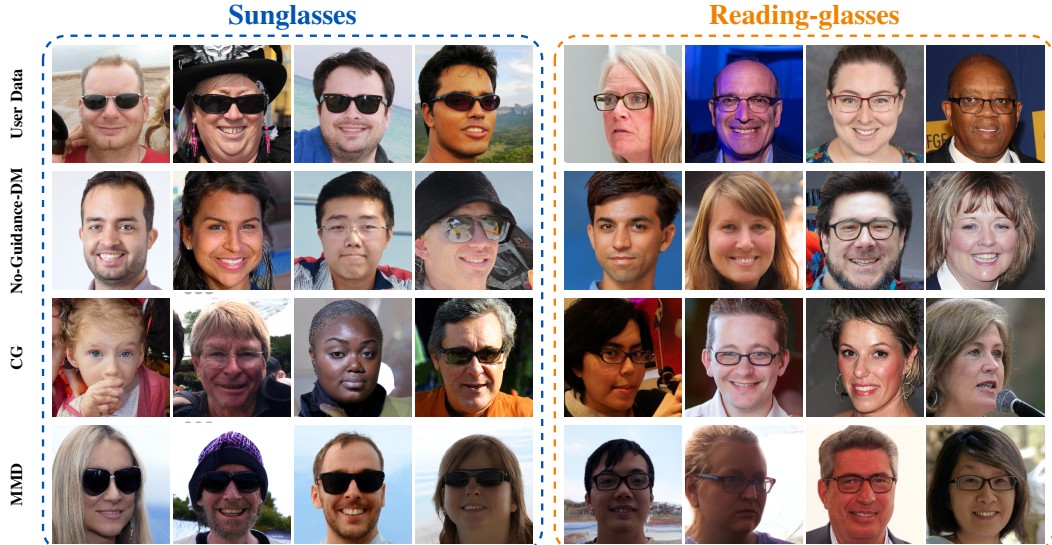

Figure 4: User's samples and generated data by unguided/guided LDMs on the FFHQ dataset.

Table 1: Evaluation scores for samples generated on FFHQ using MMD guidance vs. baselines.

| User | Guidance | FD ↓ | KD ↓ | RRKE ↓ | Density ($\times 10^2$) ↑ | Coverage ($\times 10^2$) ↑ |
|------|----------|------|------|--------|--------|----------|
| Sunglasses | No-Guidance-DM | $1220.51 \pm 131.78$ | $8.21 \pm 1.79$ | $1.73 \pm 0.16$ | $74.18 \pm 23.43$ | $30.88 \pm 11.48$ |
| | User-Trained-DM | $1004.71 \pm 51.23$ | $8.50 \pm 0.45$ | $1.52 \pm 0.03$ | $43.43 \pm 5.70$ | $57.96 \pm 1.60$ |
| | Fine-tuning | $747.69 \pm 59.18$ | $4.15 \pm 0.52$ | $1.40 \pm 0.02$ | $71.64 \pm 4.66$ | $71.12 \pm 2.85$ |
| | CG | $1195.85 \pm 121.26$ | $8.07 \pm 1.80$ | $1.71 \pm 0.15$ | $73.42 \pm 23.67$ | $27.24 \pm 12.20$ |
| | Domain Guidance | $710.53 \pm 30.45$ | $4.06 \pm 0.16$ | $1.43 \pm 0.06$ | $87.38 \pm 15.07$ | $75.44 \pm 2.71$ |
| | MMD (Ours) | $692.87 \pm 30.43$ | $3.25 \pm 0.18$ | $1.39 \pm 0.04$ | $113.13 \pm 9.36$ | $79.08 \pm 1.54$ |
| Reading-glasses | No-Guidance-DM | $702.83 \pm 27.10$ | $2.22 \pm 0.16$ | $1.35 \pm 0.01$ | $68.79 \pm 13.60$ | $80.96 \pm 2.50$ |
| | User-Trained-DM | $1105.12 \pm 53.03$ | $7.54 \pm 0.29$ | $1.47 \pm 0.01$ | $59.14 \pm 4.18$ | $57.52 \pm 2.91$ |
| | Fine-tuning | $732.91 \pm 43.02$ | $2.99 \pm 0.20$ | $1.35 \pm 0.01$ | $82.32 \pm 4.47$ | $73.40 \pm 1.89$ |
| | CG | $678.48 \pm 21.16$ | $2.16 \pm 0.14$ | $1.34 \pm 0.01$ | $70.12 \pm 12.81$ | $77.96 \pm 1.28$ |
| | Domain Guidance | $667.56 \pm 21.04$ | $2.08 \pm 0.13$ | $1.34 \pm 0.01$ | $75.74 \pm 14.97$ | $81.72 \pm 2.30$ |
| | MMD (Ours) | $574.29 \pm 17.57$ | $1.39 \pm 0.05$ | $1.30 \pm 0.01$ | $87.10 \pm 9.69$ | $84.60 \pm 2.54$ |

Table 2: Evaluation metrics for the synthetic dataset used in Figure 1. Comparison of SDXL and PixArt with No-Guidance-DM vs. MMD Guidance.

| Model | Guidance | FD ↓ | KD ↓ | RRKE ↓ | Density ($\times 10^2$) ↑ | Coverage ($\times 10^2$) ↑ |
|-------|----------|------|------|--------|--------|----------|
| SDXL | No-Guidance-DM | 1953.75 | 3.57 | 1.93 | 5.63 | 11.34 |
| | MMD (Ours) | 1674.45 | 2.49 | 1.79 | 18.01 | 34.20 |
| PixArt | No-Guidance-DM | 1358.40 | 0.80 | 1.34 | 55.32 | 64.29 |
| | MMD (Ours) | 1060.68 | 0.66 | 1.27 | 67.80 | 75.34 |

Table 3: Runtime analysis of MMD guidance vs. no guidance diffusion sample generation.

| Model | Guidance | #Samples generated | | | | | | | | | |
|-------|----------|----|-----|-----|-----|-----|-----|-----|-----|-----|-----|
| | | 50 | 100 | 150 | 200 | 250 | 300 | 350 | 400 | 450 | 500 |
| SDXL | No-Guidance-DM | 328 | 634 | 946 | 1258 | 1603 | 1914 | 2208 | 2516 | 2832 | 3142 |
| | MMD (Ours) | 356 | 720 | 1036 | 1378 | 1722 | 2069 | 2404 | 2749 | 3112 | 3447 |
| PixArt | No-Guidance-DM | 317 | 632 | 927 | 1282 | 1607 | 1852 | 2136 | 2435 | 2771 | 3063 |
| | MMD (Ours) | 342 | 683 | 1018 | 1348 | 1685 | 2024 | 2352 | 2694 | 3019 | 3356 |

reference dataset was produced with SD-XL. We then evaluated MMD Guidance by sampling from the diffusion model without explicitly specifying the styles. As shown in Figure 1 and Figure 19 in Appendix C, the guided model successfully reproduced the visual characteristics of the target distribution, while the unguided LDM produced samples with mismatched styles. Numerical scores in Table 2 confirm this observation: RRKE, FD, and KD decreased, suggesting improved distributional alignment, while Coverage increased, indicating improved diversity. To further evaluate

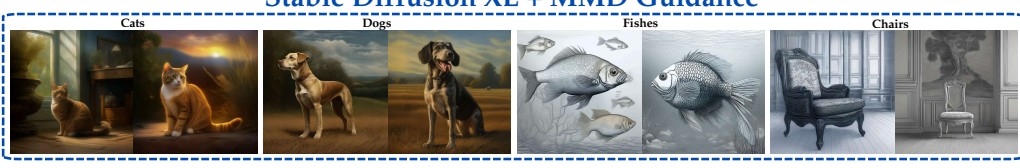

Figure 5: Qualitative comparison of reference set and MMD-guided image generation with SDXL.

Table 4: Evaluation metrics as a function of the number of reference samples.

| # of reference samples | Guidance | FD $\downarrow$ | KD $\downarrow$ | RRKE $\downarrow$ |
|---|---|---|---|---|
| $n = 0$ | No Guidance | 1953.75 | 3.57 | 1.93 |
| $n = 10$ | MMD | 1805.92 | 2.7657 | 1.89 |
| $n = 50$ | MMD | 1731.27 | 2.6882 | 1.83 |
| $n = 100$ | MMD | 1691.42 | 2.5682 | 1.81 |
| $n = 150$ | MMD | 1686.61 | 2.5301 | 1.80 |
| $n = 200$ | MMD | 1674.45 | 2.4945 | 1.79 |

MMD guidance on prompt-aware LDMs, we extended our experiments by generating eight categories of animals and objects, each rendered in four distinct styles (10,000 images in total). For each run, four categories were randomly selected, with one style assigned to each. Qualitative results are shown in Figure 5, where the MMD-guided SDXL samples closely match the reference styles. Additional results are provided in Appendix C.

**Scalability and time complexity of prompt-aware MMD guidance.** To evaluate the time complexity of our proposed MMD guidance for LDMs, in Table 3, we report the cumulative time for generating a group of $n$ samples with and without MMD guidance for large-scale prompt-aware LDMs using a 50-step diffusion process on an NVIDIA RTX 4090. The time values reported in the table support the computational efficiency of MMD guidance in the latent space of LDMs.

**Effect of the number of reference samples on the MMD Guidance.** we have measured the norm difference of the MMD gradient estimated with $N_r$ samples (empirical) and with 10000 samples (population estimate), and report the average $L_2$-norm errors over generating 1000 samples. As can be seen, the estimate becomes highly accurate, even with 100 data points, which follows the choice of MMD distance with the Gaussian and polynomial degree-3 kernels in our experiments.

**Comparing MMD with Domain Guidance baseline.** We compare the effectiveness of our method against the Domain Guidance baseline (Zhong et al., 2025) on the FFHQ dataset; here, guidance is applied toward two user-specific subsets, each containing 500 images: one subset consists of images of people wearing sunglasses, and the other wearing reading glasses. As Tables 1 indicate, the MMD guidance consistently outperforms the Domain Guidance baseline across the reported metrics.

## 7 CONCLUSION

We presented MMD Guidance, a training-free adaptation method that enables diffusion models to generate samples matching arbitrary target distributions specified through small reference sets. By augmenting the reverse diffusion process with gradients of the Maximum Mean Discrepancy, our approach achieves distribution alignment without modifying model parameters, a useful capability when computational and data resources for retraining are limited. While MMD Guidance performs well with a limited number of reference samples, extremely sparse scenarios with highly limited data remain challenging due to variance in gradient estimation. The choice of kernel function may impact the performance, and adaptive kernel selection could further improve robustness. Future work could explore combining MMD with other divergences to leverage their complementary strengths, and also extending the framework to video generation, where sequential consistency adds to the complexity.

## REPRODUCIBILITY STATEMENT

We have implemented several measures to facilitate the reproducibility of our work. All theoretical results are stated with assumptions and accompanied by proofs in Appendix A. The proposed algorithms (Algorithms 1 and 2) are presented in detail, including their guidance update rules, to enable re-implementation. Experimental settings, datasets, evaluation metrics, and runtime analyses are described in Section 6 and Appendices B,C. To further facilitate reproducibility, we will release an anonymous code for MMD Guidance as part of the supplementary materials.

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

# A PROOFS

## A.1 PROOF OF THEOREM 1

Define the centered random vectors:

$$Y_j = \nabla_{z_0} k(z_0, z_j^{(r)}) - \mathbb{E}_{z' \sim Q}[\nabla_{z_0} k(z_0, z')], \quad j = 1, \dots, N_r. \tag{13}$$

Since the reference samples $z_j^{(r)}$ are drawn i.i.d. from $Q$ independently of how $z_0$ was generated, the $Y_j$ are independent random vectors in $\mathbb{R}^d$ with $\mathbb{E}[Y_j] = 0$ (following its definition) and $\|Y_j\|_2 \leq 2L$ almost surely, since $\|\nabla_{z_0} k(z_0, \cdot)\|_2 \leq L$ by the Lipschitz property.

Applying the Hoeffding inequality for random Hilbert-Schmidt operators (Sutherland et al., 2018), we have the following to hold with probability at least $1 - \delta$:

$$\left\| \frac{1}{N_r} \sum_{j=1}^{N_r} Y_j \right\|_2 \leq \frac{2L}{\sqrt{N_r}} \left( 1 + \sqrt{2 \log(\frac{1}{\delta})} \right). \tag{14}$$

Since $\widehat{g}_{\text{cross}}(z_0) - g^*_{\text{cross}}(z_0) = -\frac{2}{N_r} \sum_{j=1}^{N_r} Y_j$, the following will hold with probability at least $1 - \delta$:

$$\left\| \widehat{g}_{\text{cross}}(z_0) - g^*_{\text{cross}}(z_0) \right\|_2 \leq \frac{4L}{\sqrt{N_r}} \left( 1 + \sqrt{2 \log(1/\delta)} \right). \tag{15}$$

## A.2 PROOF OF COROLLARY 1

For the Gaussian RBF kernel, the gradient is $\nabla_x k(x, y) = -\frac{1}{\sigma^2} k(x, y)(x - y)$, yielding

$$\left\| \nabla_x k(x, y) \right\|_2 = \frac{\|x - y\|_2}{\sigma^2} \exp\left(-\frac{\|x - y\|_2^2}{2\sigma^2}\right)$$

Setting $t = \|x - y\|_2 / (\sigma \sqrt{2})$, this becomes $(\sqrt{2}/\sigma) t e^{-t^2}$. Since $\max_{t \geq 0} t e^{-t^2} = e^{-1/2}/\sqrt{2}$ (achieved at $t = 1/\sqrt{2}$), we have the Lipschitz constant $L = 1/(\sigma \sqrt{e})$. Knowing that $\frac{4}{\sqrt{e}} < 3$, we can plug this Lipschitz constant into the assumption of Theorem 1, which yields the result.

## A.3 PROOF OF THEOREM 2

The theorem follows from applying the following covering-number bound to the result of Theorem 1. Consider a positive $\varepsilon > 0$, and let $\mathcal{C}_\varepsilon$ be an $\varepsilon$-covering net of $\mathcal{Z}$ in the $\ell_2$ metric. Every $z \in \mathcal{Z}$ has a representative $z' \in \mathcal{C}_\varepsilon$ with $\|z - z'\|_2 \leq \varepsilon$. For a Euclidean ball of radius $R$, a well-known covering number bound of the $\ell_2$-ball is $|\mathcal{C}_\varepsilon| \leq (6R/\varepsilon)^d$.

For a fixed $z \in \mathcal{C}_\varepsilon$, Theorem 1 provides the following pointwise bound to hold with probability at least $1 - \eta$:

$$\|\widehat{g}_{\text{cross}}(z) - g^*_{\text{cross}}(z)\|_2 \leq \frac{4L}{\sqrt{N_r}} \left( 1 + \sqrt{2 \log \frac{1}{\eta}} \right)$$

Choosing $\eta = \delta / |\mathcal{C}_\varepsilon|$ and applying the union bound across all $z \in \mathcal{C}_\varepsilon$, we conclude that with probability at least $1 - \delta$,

$$\max_{z \in \mathcal{C}_\varepsilon} \|\widehat{g}_{\text{cross}}(z) - g^*_{\text{cross}}(z)\|_2 \leq \frac{4L}{\sqrt{N_r}} \left( 1 + \sqrt{2 \log \frac{|\mathcal{C}_\varepsilon|}{\delta}} \right).$$

To extend this guarantee from the net to the entire ball, consider an arbitrary $z_0 \in \mathcal{Z}$ and its nearest covering net point $z' \in \mathcal{C}_\varepsilon$ which by definition satisfies $\|z_0 - z'\|_2 \leq \varepsilon$. The deviation can be decomposed as follows using the triangle inequality:

$$\left\| \widehat{g}_{\text{cross}}(z_0) - g^*_{\text{cross}}(z_0) \right\|_2$$
$$\leq \left\| \widehat{g}_{\text{cross}}(z_0) - \widehat{g}_{\text{cross}}(z') \right\|_2 + \left\| \widehat{g}_{\text{cross}}(z') - g^*_{\text{cross}}(z') \right\|_2 + \left\| g^*_{\text{cross}}(z') - g^*_{\text{cross}}(z_0) \right\|_2.$$

The middle term is controlled by the union bound over the net. The first and last terms follows from the Lipschitz constant of $\nabla k$. Thus,

$$\|\widehat{g}_{\text{cross}}(z_0) - \widehat{g}_{\text{cross}}(z')\|_2 = \left\| \frac{2}{N_r} \sum_{j=1}^{N_r} \left( \nabla_{z_0} k(z_0, z_j^{(r)}) - \nabla_{z'} k(z', z_j^{(r)}) \right) \right\|_2$$

$$\leq \frac{2}{N_r} \sum_{j=1}^{N_r} L' \|z_0 - z'\|_2$$

$$\leq 2L'\varepsilon,$$

and similarly $\|g_{\text{cross}}^*(z') - g_{\text{cross}}^*(z_0)\|_2 \leq 2L'\varepsilon$. Combining these inequalities, we obtain

$$\left\| \widehat{g}_{\text{cross}}(z_0) - g_{\text{cross}}^*(z_0) \right\|_2 \leq \frac{4L}{\sqrt{N_r}} \left( 1 + \sqrt{2 \log \frac{|\mathcal{C}_\varepsilon|}{\delta}} \right) + 4L'\varepsilon.$$

Substituting $|\mathcal{C}_\varepsilon| \leq (6R/\varepsilon)^d$ and choosing $\varepsilon = 1/\sqrt{N_r}$ completes the bound. The proof is therefore complete.

## A.4 PROOF OF COROLLARY 2

**Corollary 2.** *For the Gaussian RBF kernel $k(x, y) = \exp(-\|x - y\|_2^2/(2\sigma^2))$ on $\mathcal{Z} = \{z : \|z\|_2 \leq R\}$, with probability at least $1 - \delta$,*

$$\sup_{z \in \mathcal{Z}} \|\widehat{g}_{\text{cross}}(z) - g_{\text{cross}}^*(z)\|_2 \leq \frac{24}{\sigma\sqrt{N_r}} \left( 1 + \sqrt{d \log \frac{R\sqrt{N_r}}{\sigma} + \log \frac{1}{\delta}} \right)$$

*Proof.* For Gaussian RBF, $\|\nabla_x k(x, y)\|_2 = \frac{\|x - y\|_2}{\sigma^2} k(x, y) \leq 1/(\sigma\sqrt{e})$, giving $L = 1/(\sigma\sqrt{e})$. The Hessian is
$$\nabla_x^2 k(x, y) = \left( \frac{(x-y)(x-y)^\top}{\sigma^4} - \frac{I}{\sigma^2} \right) k(x, y),$$

whose operator norm is maximized when $\|x - y\|_2^2 = \sigma^2$, leading to $\|\nabla_x^2 k(x, y)\|_{\text{op}} \leq 2/(\sigma^2\sqrt{e})$. Thus $L' = 2/(\sigma^2\sqrt{e})$. Substituting into Theorem 2's final inequality gives the bound in the corollary. $\square$

## A.5 PROOF OF THEOREM 3

**Theorem 3** (Concentration for Weighted Cross Term in Product Kernel). *Let $\mathcal{P} \subseteq \mathbb{R}^{d_p}$ and $\mathcal{Z} \subseteq \mathbb{R}^{d_z}$ be compact. Let $k_p : \mathcal{P} \times \mathcal{P} \to [-1, 1]$ and $k_z : \mathcal{Z} \times \mathcal{Z} \to [-1, 1]$ be normalized kernels with $k_p(p, p) = k_z(z, z) = 1$. Assume $k_z$ is $L_z$-Lipschitz continuous in its first argument. Let $Q \times \Pi'$ be the target joint distribution and let $\{(p_j^{(r)}, z_j^{(r)})\}_{j=1}^{N_r} \overset{iid}{\sim} Q \times \Pi'$ be reference samples. For a pair $(p_0, z_0) \in \mathcal{P} \times \mathcal{Z}$, we define*

$$g_{cross,\otimes}^*(p_0, z_0) = -2\mathbb{E}_{(p', z') \sim Q \times \Pi'}[k_p(p_0, p') \nabla_{z_0} k_z(z_0, z')],$$

$$\widehat{g}_{cross,\otimes}(p_0, z_0) = -\frac{2}{N_r} \sum_{j=1}^{N_r} k_p(p_0, p_j^{(r)}) \nabla_{z_0} k_z(z_0, z_j^{(r)}).$$

*Then for any $\delta \in (0, 1)$, with probability at least $1 - \delta$ over the draw of reference samples:*

$$\left\| \widehat{g}_{cross,\otimes}(p_0, z_0) - g_{cross,\otimes}^*(p_0, z_0) \right\|_2 \leq \frac{4L_z}{\sqrt{N_r}} \left( 1 + \sqrt{2 \log(1/\delta)} \right).$$

*Proof.* We define the centered random vectors

$$W_j = k_p(p_0, p_j^{(r)}) \nabla_{z_0} k_z(z_0, z_j^{(r)}) - \mathbb{E}_{(p', z') \sim Q \times \Pi'}[k_p(p_0, p') \nabla_{z_0} k_z(z_0, z')], \quad j = 1, \ldots, N_r. \tag{16}$$

Since the reference pairs $(p_j^{(r)}, z_j^{(r)})$ are drawn i.i.d. from $Q \times \Pi'$ independently of how $(p_0, z_0)$ was generated, the $W_j$ are independent random vectors in $\mathbb{R}^{d_z}$ with $\mathbb{E}[W_j] = 0$ (by construction) and $\|W_j\|_2 \leq 2L_z$ almost surely, since $k_p \in [-1, 1]$ and $\|\nabla_{z_0} k_z(z_0, \cdot)\|_2 \leq L_z$.

Applying the Hoeffding inequality for Hilbert-Schmidth operators (Sutherland et al., 2018), we have the following with probability at least $1 - \delta$:

$$\left\| \frac{1}{N_r} \sum_{j=1}^{N_r} W_j \right\|_2 \leq \frac{2L_z}{\sqrt{N_r}} \left( 1 + \sqrt{2 \log(1/\delta)} \right). \tag{17}$$

Since $\widehat{g}_{\text{cross},\otimes}(p_0, z_0) - g^*_{\text{cross},\otimes}(p_0, z_0) = -\frac{2}{N_r} \sum_{j=1}^{N_r} W_j$, the above inequality will result in the following to hodl with probability at least $1 - \delta$:

$$\left\| \widehat{g}_{\text{cross},\otimes}(p_0, z_0) - g^*_{\text{cross},\otimes}(p_0, z_0) \right\|_2 \leq \frac{4L_z}{\sqrt{N_r}} \left( 1 + \sqrt{2 \log(1/\delta)} \right). \tag{18}$$

$\square$

**Corollary 3** (Product Kernel with Gaussian RBF). *For the product kernel $k_\otimes((p, z), (p', z')) = k_p(p, p') \cdot k_z(z, z')$ where $k_z$ is the Gaussian RBF kernel with bandwidth $\sigma$ and $k_p \in [-1, 1]$, the weighted cross term satisfies with probability at least $1 - \delta$:*

$$\left\| \widehat{g}_{cross,\otimes}(p_0, z_0) - g^*_{cross,\otimes}(p_0, z_0) \right\|_2 \leq \frac{3}{\sigma\sqrt{N_r}} \left( 1 + \sqrt{2 \log(1/\delta)} \right). \tag{19}$$

*Proof.* From Corollary 1, the Gaussian RBF kernel has Lipschitz constant $L_z = 1/(\sigma\sqrt{e})$. Applying Theorem 3 directly gives the stated bound. $\square$

# B  ALGORITHMIC DESCRIPTIONS

Here, we present Algorithm 2 for prompt-conditioned MMD guidance of diffusion models. We recall that given generated pairs $\{(p_i, z_t^{(i)})\}_{i=1}^B$ with distribution $\widehat{P}_t$ and reference pairs $\{(p_j^{(r)}, z_j^{(r)})\}_{j=1}^{N_r}$ with distribution $\widehat{Q}$, the empirical squared MMD in the product space for gradient calculation is:

$$\widehat{\text{MMD}}_\otimes^2(\widehat{P}_t, \widehat{Q}) = \frac{1}{B^2} \sum_{i=1}^B \sum_{i'=1}^B k_p(p_i, p_{i'}) k_z(z_t^{(i)}, z_t^{(i')}) + \frac{1}{N_r^2} \sum_{j=1}^{N_r} \sum_{j'=1}^{N_r} k_p(p_j^{(r)}, p_{j'}^{(r)}) k_z(z_j^{(r)}, z_{j'}^{(r)})$$

$$- \frac{2}{BN_r} \sum_{i=1}^B \sum_{j=1}^{N_r} k_p(p_i, p_j^{(r)}) k_z(z_t^{(i)}, z_j^{(r)}). \tag{20}$$

# C  ADDITIONAL NUMERICAL RESULTS

In this section, we provide additional numerical results for the MMD Guidance method. For all experiments the MMD Guidance scale is chosen via grid search on a held-out validation split using FD. For all non-text GMM guidance experiments on real images, we used MMD with cubic polynomial kernel.

**Experiment settings on GMM.** We evaluated the methods listed in Table 5 in the following setting. We reported results on three users that each has three random components (similar to Figure 7). Each component in a user has 50 data points. We applied MMD with an RBF kernel, using bandwidth 1 for the 8-component GMM and bandwidth 4 for the 25-component GMM (since it is more aligned with data from the Gaussian mixtures). The guidance scale is in order of $10^{-1}$. For the CG baseline, we trained a linear classifier (single FC layer).

**Additional results on synthetic data.** Similar to Figure 2 we compare our method to the baselines with different number of components for users, in Figures 7, and 8. Additionally, we evaluated

**Algorithm 2:** Prompt-Aware MMD-Guided Sampling

**Input:** Reference pairs $\{(p_j^{(r)}, x_j^{(r)})\}_{j=1}^{N_r}$, prompts $\{\text{prompt}_i\}_{i=1}^{B}$, guidance schedule $\{\lambda_t\}_{t=1}^{T}$

**Output:** Generated samples $\{x^{(i)}\}_{i=1}^{B}$ matching prompts and reference style

*Preprocessing:*

    $z_j^{(r)} \leftarrow \mathcal{E}(x_j^{(r)})$ for all $j \in [N_r]$

    $p_i \leftarrow \text{CLIP}_{\text{text}}(\text{prompt}_i)$ for all $i \in [B]$

    Compute $K_{ij} = k_p(p_i, p_j^{(r)})$ for all $i \in [B], j \in [N_r]$

*Initialization:* $z_T^{(i)} \sim \mathcal{N}(0, I)$ for all $i \in [B]$;

**for** $t = T$ **to** 1 **do**

    **for** $i = 1$ **to** $B$ *in parallel* **do**

        $\hat{z}_{t-1}^{(i)} \leftarrow \text{sampler}(z_t^{(i)}, t, \epsilon_\theta, p_i)$

        $g^{(i)} \leftarrow \nabla_{z_t^{(i)}} \widehat{\text{MMD}}_\otimes^2$ using precomputed $K_{ij}$ // `Using equation 12`

        $z_{t-1}^{(i)} \leftarrow \hat{z}_{t-1}^{(i)} - \lambda_t g^{(i)}$;

**return** $\{x^{(i)} = \mathcal{D}(z_0^{(i)})\}_{i=1}^{B}$

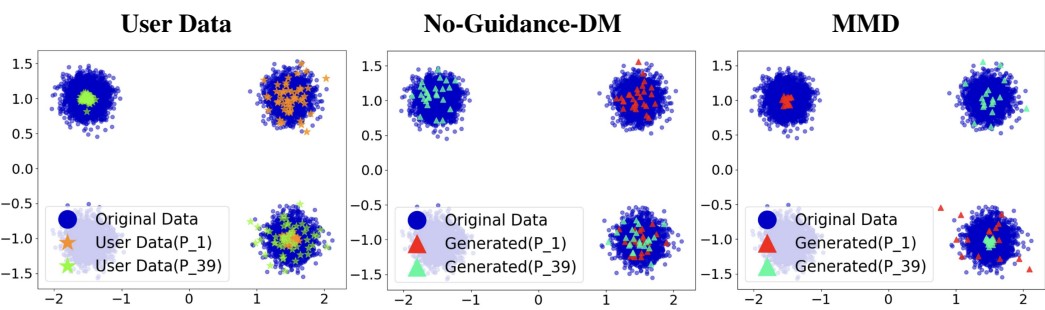

Figure 6: Comparison of MMD Guidance with baselines on 100-D Gaussian distributions, when guiding toward a user with 3 Gaussian components.

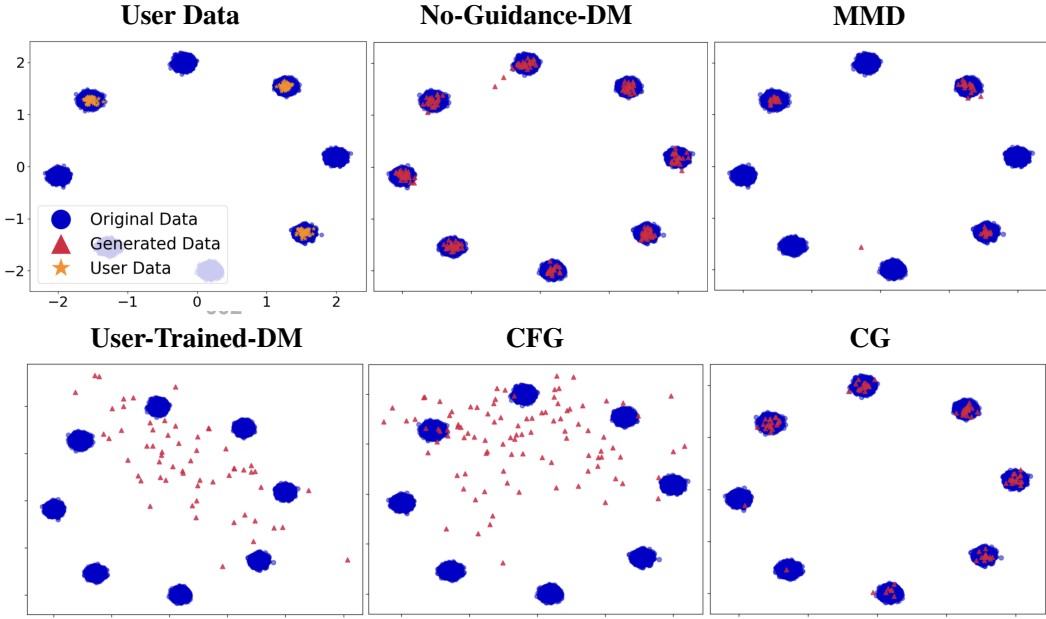

Figure 7: Comparison of MMD Guidance with baselines on 100D Gaussian distributions, when guiding toward a user with 3 Gaussian components.

metrics on three users under 8-component and 25-component GMM settings, computing all metrics

Table 5: Evaluation metrics for samples generated from three users, using 8, and 25 Gaussian component synthetic data.

| User | Guidance | 8 Gaussian Component | | | 25 Gaussian Component | | |
|---|---|---|---|---|---|---|---|
| | | FD ↓ | KD($\times 10^3$) ↓ | RRKE ↓ | FD ↓ | KD($\times 10^3$) ↓ | RRKE ↓ |
| User 1 | No-Guidance-DM | $7.04 \pm 0.002$ | $142.42 \pm 0.07$ | $2.314 \pm 0.001$ | $77.70 \pm 0.015$ | $330.91 \pm 0.28$ | $2.073 \pm 0.001$ |
| | User-Trained-DM | $27.90 \pm 0.009$ | $219.64 \pm 0.14$ | $64.135 \pm 0.103$ | $47.36 \pm 0.026$ | $260.39 \pm 0.35$ | $1.803 \pm 0.000$ |
| | Fine-tuning | $0.48 \pm 0.001$ | $27.86 \pm 0.10$ | $1.051 \pm 0.001$ | $4.72 \pm 0.004$ | $16.65 \pm 0.11$ | $1.290 \pm 0.000$ |
| | CG | $0.94 \pm 0.002$ | $24.13 \pm 0.10$ | $1.219 \pm 0.001$ | $32.68 \pm 0.006$ | $81.31 \pm 0.17$ | $0.398 \pm 0.000$ |
| | CFG | $0.97 \pm 0.001$ | $136.06 \pm 0.11$ | $4.833 \pm 0.003$ | $7.79 \pm 0.007$ | $134.12 \pm 0.31$ | $0.270 \pm 0.000$ |
| | MMD (Ours) | $0.37 \pm 0.001$ | $17.02 \pm 0.08$ | $1.294 \pm 0.002$ | $3.56 \pm 0.003$ | $9.52 \pm 0.07$ | $0.231 \pm 0.000$ |
| User 2 | No-Guidance-DM | $7.46 \pm 0.004$ | $153.98 \pm 0.16$ | $2.089 \pm 0.003$ | $95.58 \pm 0.034$ | $417.04 \pm 0.65$ | $2.356 \pm 0.000$ |
| | User-Trained-DM | $32.04 \pm 0.008$ | $217.24 \pm 0.24$ | $76.478 \pm 0.037$ | $88.33 \pm 0.020$ | $260.85 \pm 0.49$ | $2.411 \pm 0.001$ |
| | Fine-tuning | $1.06 \pm 0.002$ | $40.19 \pm 0.10$ | $1.231 \pm 0.001$ | $33.97 \pm 0.022$ | $135.06 \pm 0.36$ | $1.557 \pm 0.000$ |
| | CG | $1.12 \pm 0.001$ | $29.23 \pm 0.09$ | $1.192 \pm 0.002$ | $41.79 \pm 0.015$ | $122.15 \pm 0.38$ | $0.482 \pm 0.000$ |
| | CFG | $1.76 \pm 0.002$ | $194.16 \pm 0.24$ | $11.096 \pm 0.009$ | $12.56 \pm 0.015$ | $152.69 \pm 0.43$ | $0.341 \pm 0.000$ |
| | MMD (Ours) | $0.33 \pm 0.001$ | $16.54 \pm 0.06$ | $1.245 \pm 0.002$ | $2.84 \pm 0.003$ | $9.00 \pm 0.08$ | $0.246 \pm 0.000$ |
| User 3 | No-Guidance-DM | $8.30 \pm 0.004$ | $167.47 \pm 0.15$ | $2.376 \pm 0.003$ | $87.63 \pm 0.023$ | $378.38 \pm 0.31$ | $2.107 \pm 0.001$ |
| | User-Trained-DM | $39.00 \pm 0.008$ | $224.98 \pm 0.20$ | $80.293 \pm 0.034$ | $78.56 \pm 0.022$ | $284.06 \pm 0.28$ | $2.362 \pm 0.001$ |
| | Fine-tuning | $1.11 \pm 0.002$ | $41.06 \pm 0.07$ | $1.295 \pm 0.002$ | $6.97 \pm 0.008$ | $26.66 \pm 0.14$ | $1.263 \pm 0.000$ |
| | CG | $1.24 \pm 0.002$ | $32.59 \pm 0.11$ | $1.110 \pm 0.001$ | $71.95 \pm 0.015$ | $407.94 \pm 0.41$ | $0.531 \pm 0.000$ |
| | CFG | $2.20 \pm 0.002$ | $197.11 \pm 0.19$ | $12.831 \pm 0.011$ | $9.49 \pm 0.010$ | $148.67 \pm 0.37$ | $0.257 \pm 0.000$ |
| | MMD (Ours) | $0.64 \pm 0.001$ | $14.46 \pm 0.09$ | $1.258 \pm 0.001$ | $1.85 \pm 0.004$ | $7.65 \pm 0.09$ | $0.236 \pm 0.000$ |

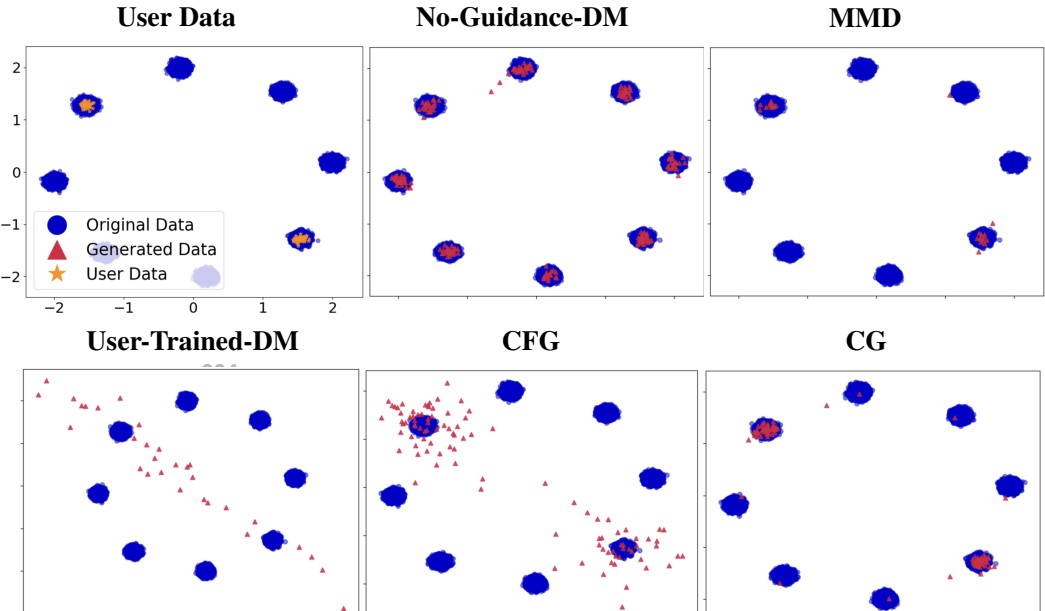

Figure 8: Comparison of MMD Guidance with baselines on 100D Gaussian distributions, when guiding toward a user with 2 Gaussian components.

on 1,000 generated samples per method. As shown in Table 5, across both mixture complexities, the MMD Guidance consistently achieves the lowest FD, KD, and RRKE in most cases. Figure 9 illustrates the effectiveness of our method in GMM settings of 8 and 25 components.

**Prompt-aware GMM. Prompt-aware synthetic GMM.** To simulate the text–style experiment in a controlled setting with known ground truth, we considered a prompt-conditioned GMM, in which the prompt corresponds to the number of the Gaussian component in the GMM. In this setup, a user provide few-shot references for selected prompts; for each prompt, the user samples share the same component mean but have different variances, simulating style shifts. As shown in Figure 6, the guided sampler allocates mass to the correct components for the queried prompts and respects their local geometry, component variance, and also preserves the target mixture ratios across components. In comparison, the unguided model only generates samples with respect to the means, ignores variance differences, and does not follow the intended proportions.

Also, we generated the results in Figure 6 under the following setup. We construct a synthetic user with data–prompt pairs drawn from a four-component GMM. The user distribution includes prompts

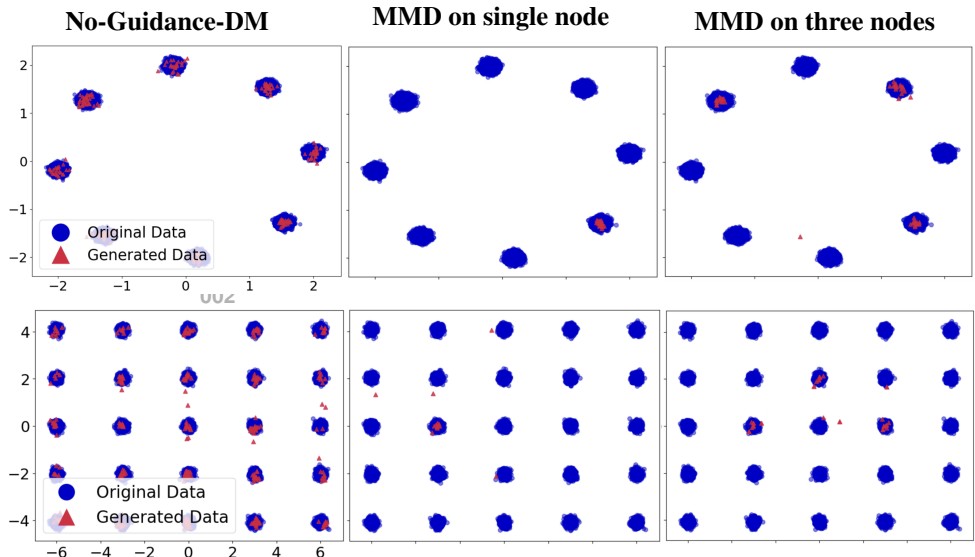

Figure 9: Effect of MMD on Gaussian mixture models with 8 and 25 components. The first row shows results for 8 components; the second row shows 25 components. Column labels denote unguided diffusion and two MMD guidance targets.

Table 6: Evaluation metrics for samples generated on FFHQ for local training and fine-tuning.

| User | Guidance | FD $\downarrow$ | KD $\downarrow$ | RRKE $\downarrow$ | Density ($\times 10^2$) $\uparrow$ | Coverage ($\times 10^2$) $\uparrow$ |
|------|----------|-----|-----|------|------------------|-------------------|
| Sunglasses | User-Trained-DM | $1004.71 \pm 51.23$ | $8.50 \pm 0.45$ | $1.52 \pm 0.03$ | $43.43 \pm 5.70$ | $57.96 \pm 1.60$ |
| | Fine-tuning | $747.69 \pm 59.18$ | $4.15 \pm 0.52$ | $1.40 \pm 0.02$ | $71.64 \pm 4.66$ | $71.12 \pm 2.85$ |
| | MMD (Ours) | $692.87 \pm 30.43$ | $3.25 \pm 0.18$ | $1.39 \pm 0.04$ | $113.13 \pm 9.36$ | $79.08 \pm 1.54$ |
| Reading-glasses | User-Trained-DM | $1105.12 \pm 53.03$ | $7.54 \pm 0.29$ | $1.47 \pm 0.01$ | $59.14 \pm 4.18$ | $57.52 \pm 2.91$ |
| | Fine-tuning | $732.91 \pm 43.02$ | $2.99 \pm 0.20$ | $1.35 \pm 0.01$ | $82.32 \pm 4.47$ | $73.40 \pm 1.89$ |
| | MMD (Ours) | $574.29 \pm 17.57$ | $1.39 \pm 0.05$ | $1.30 \pm 0.01$ | $87.10 \pm 9.69$ | $84.60 \pm 2.54$ |

"one" and "thirty nine", but with a user-specific target variance within the corresponding mixture component. As shown in Figure 6, MMD Guidance reproduces both the correct components and the target variance (style), where the No-Guidance-DM model selects the right component but does not match the variance.

**Experiment settings on FFHQ.** We evaluated the methods listed in Table 1 using the following setup: Each user contains 500 images from the FFHQ dataset. (i) a user who has images of people wearing sunglasses; (ii) a user whose images predominantly feature people wearing reading-glasses. The user images were passed through the LDM AutoEncoder to construct the features used for the MMD Guidance in latent space. In FFHQ experiments, we generated samples using pretrained LDM weights and a DDPM sampler. The MMD Guidance scale is in order of $10^{-4}$, and all metrics are computed on 500 generated samples per method. For the CG baseline, we trained a classifier with an Inception v3Szegedy et al. (2016) backbone and a linear classification head.

**Additional results on FFHQ.** We also compare our method to baselines that require additional user-specific training (User-Trained-DM and fine-tuning). Table 6 shows that, across both users, the MMD Guidance outperforms these baselines. Beyond superior results, MMD Guidance is substantially more efficient in both time and compute, as it does not require any per-user training. In practice, the additional cost is negligible compared to unguided sampling (more details in Runtime Analysis). Furthermore, since these baselines are trained on a very small per-user subset (500 images), the diffusion model is prone to overfitting; in particular, the User-Trained-DM baseline shows clear memorization of the training data, as illustrated in Figure 10.

**Experiment settings on CelebA-HQ.** We conducted experiments on two users derived from a CelebA-HQ subset to capture specific styles: one with 500 images of people with blond hair and one with 500 images of people with black hair. The user images were passed through the LDM AutoEncoder to construct the features used for the MMD Guidance on latent space. For CelebA-HQ experiments, we generated samples using pretrained LDM weights and a DDPM sampler. The

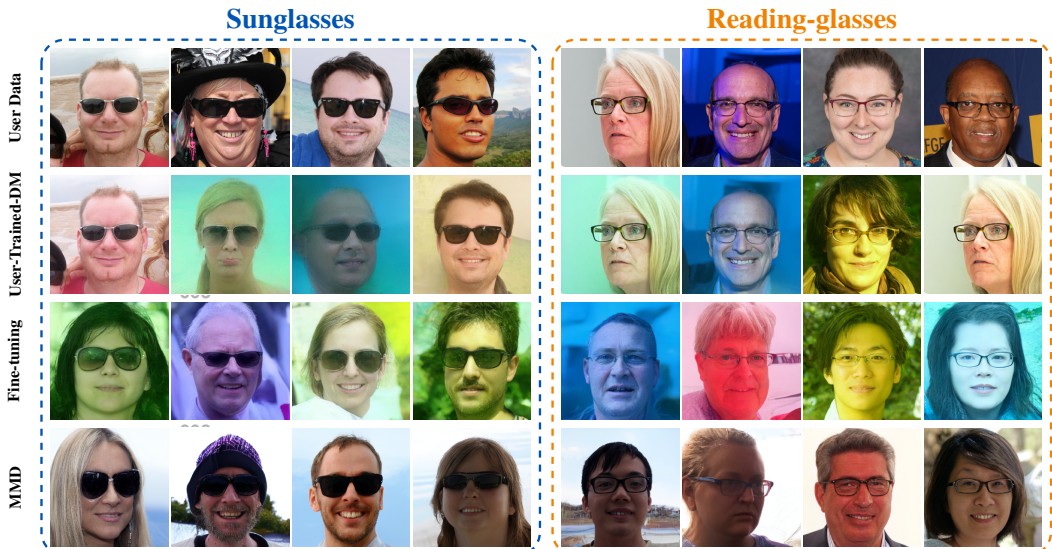

Figure 10: Comparison of MMD Guidance with training baselines on FFHQ dataset.

Table 7: Comparison metrics for samples generated on CelebA-HQ using MMD versus the baselines.

| User | Guidance | FD ↓ | KD ↓ | RRKE ↓ | Density ($\times 10^2$) ↑ | Coverage ($\times 10^2$) ↑ |
|---|---|---|---|---|---|---|
| Black-haired | No-Guidance-DM | $665.99 \pm 22.26$ | $2.61 \pm 0.38$ | $1.32 \pm 0.01$ | $94.68 \pm 4.55$ | $83.62 \pm 6.96$ |
| | User-Trained-DM | $921.55 \pm 18.76$ | $5.70 \pm 0.39$ | $1.41 \pm 0.01$ | $86.16 \pm 7.95$ | $70.09 \pm 3.31$ |
| | Fine-tuning | $817.17 \pm 14.97$ | $4.86 \pm 0.40$ | $1.37 \pm 0.01$ | $89.38 \pm 1.26$ | $75.26 \pm 4.40$ |
| | CG | $628.59 \pm 14.52$ | $2.18 \pm 0.30$ | $1.30 \pm 0.01$ | $108.02 \pm 5.76$ | $91.14 \pm 2.42$ |
| | MMD (Ours) | $627.57 \pm 18.20$ | $2.06 \pm 0.28$ | $1.30 \pm 0.01$ | $104.64 \pm 4.54$ | $90.75 \pm 2.95$ |
| Blond-haired | No-Guidance-DM | $668.17 \pm 24.25$ | $2.99 \pm 0.23$ | $1.44 \pm 0.02$ | $66.82 \pm 9.52$ | $82.69 \pm 4.42$ |
| | User-Trained-DM | $779.93 \pm 14.70$ | $4.75 \pm 0.21$ | $1.49 \pm 0.01$ | $60.30 \pm 8.27$ | $66.39 \pm 3.40$ |
| | Fine-tuning | $640.63 \pm 13.07$ | $4.04 \pm 0.07$ | $1.40 \pm 0.01$ | $89.50 \pm 11.32$ | $69.56 \pm 4.77$ |
| | CG | $663.36 \pm 24.96$ | $2.88 \pm 0.23$ | $1.44 \pm 0.02$ | $67.58 \pm 10.14$ | $85.05 \pm 5.16$ |
| | MMD (Ours) | $637.31 \pm 27.12$ | $2.28 \pm 0.20$ | $1.41 \pm 0.02$ | $80.98 \pm 11.56$ | $89.72 \pm 3.50$ |

MMD Guidance scale is in order of $10^{-5}$, and all metrics are computed on 200 generated samples per method. For the CG baseline, we trained a classifier with an Inception v3 Szegedy et al. (2016) backbone and a linear classification head.

**CelebA-HQ results.** We computed the evaluation metrics on CelebA-HQ and compared them with No-Guidance-DM, CG, fine-tuning, and User-Trained-DM in Table 7. Across both user groups, the MMD Guidance achieves the strongest overall performance. Similarly to FFHQ, the limited per-user data (500 images) leads the User-Trained-DM baseline to overfit (memorize user data). Qualitative comparisons are also provided in Figure 11.

**Experiment settings on Stable Diffusion v1.4.** We evaluate the methods in Table 8 under the following setting. We use Stable Diffusion v1.4 at $512 \times 512$ resolution with 50 inference steps. For each image style (winter, black-and-white, sketch, cartoon), we first generate 200 images using the prompts "bicycle in winter time", "black and white photo of a bicycle", "sketch of a bicycle", and "cartoonish bicycle", with a fixed classifier-free guidance (CFG) scale of 5. During MMD Guidance, we use the prompt "bicycle". Figure 12 illustrates the effect of MMD Guidance on the generated styles. For SD v1.4 experiments, the MMD Guidance scale is on the order of $10^{-2}$, and all metrics are computed on 500 generated samples per method. The same setting is used for the results in Table 9, except that all prompts (for both the user construction and guided generation) use "car" instead of "bicycle". The qualitative results are included in Figure 13.

**Ablation study on MMD Guidance scale and MMD kernel.** Figures 14, 15, and 16 analyze the effect of the kernel choice and the MMD Guidance scale $\alpha$ on FD and KD. The comparisons use black-and-white images from the bike-user experiments with Stable Diffusion v1.4. It is evident from these results that polynomial kernels ($d \in \{2, 3, 4\}$), and RBF kernels ($\sigma \in \{1.25, 1.5, 2\}$) follow very similar FD/KD curves, and the optimal $\alpha$ appears in the same range. At the best $\alpha$,

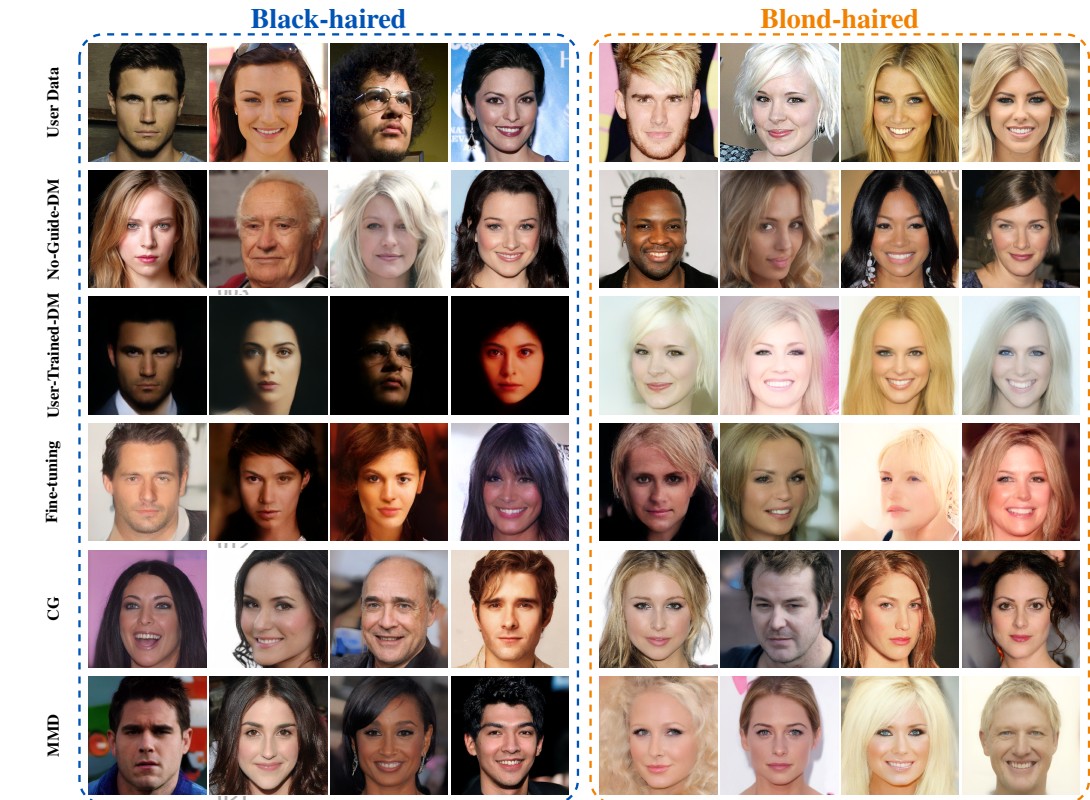

Figure 11: Comparison of MMD Guidance and baselines on CelebA-HQ dataset.

Table 8: Evaluated metrics for **Bike**. Each block shows a user (Cartoon, Winter, B/W, and Sketch) with No-Guidance-DM vs. MMD Guidance.

| User | Guidance | FD ↓ | KD ↓ | RRKE ↓ | Density ($\times 10^2$) ↑ | Coverage ($\times 10^2$) ↑ |
|------|----------|------|------|--------|---------|----------|
| B/W | No-Guidance-DM | $354.30 \pm 8.76$ | $3.83 \pm 0.19$ | $1.43 \pm 0.02$ | $43.56 \pm 3.57$ | $69.18 \pm 2.42$ |
| | MMD (Ours) | $241.40 \pm 9.30$ | $1.96 \pm 0.13$ | $1.32 \pm 0.01$ | $63.62 \pm 4.04$ | $82.22 \pm 2.98$ |
| Winter | No-Guidance-DM | $1147.48 \pm 33.88$ | $12.60 \pm 0.38$ | $1.84 \pm 0.03$ | $9.28 \pm 4.50$ | $6.45 \pm 0.63$ |
| | MMD (Ours) | $1140.07 \pm 33.84$ | $12.99 \pm 0.37$ | $1.82 \pm 0.02$ | $12.32 \pm 6.31$ | $6.73 \pm 0.88$ |
| Cartoon | No-Guidance-DM | $1583.38 \pm 9.88$ | $22.03 \pm 0.27$ | $2.56 \pm 0.04$ | $53.63 \pm 9.26$ | $6.06 \pm 0.67$ |
| | MMD (Ours) | $1516.45 \pm 11.97$ | $21.43 \pm 0.27$ | $2.54 \pm 0.04$ | $46.99 \pm 10.93$ | $8.08 \pm 0.65$ |
| Sketch | No-Guidance-DM | $949.10 \pm 7.68$ | $17.40 \pm 0.30$ | $2.53 \pm 0.06$ | $0.02 \pm 0.04$ | $0.12 \pm 0.18$ |
| | MMD (Ours) | $703.12 \pm 7.35$ | $12.42 \pm 0.24$ | $1.99 \pm 0.03$ | $2.62 \pm 0.71$ | $9.57 \pm 2.32$ |

Table 9: Evaluated metrics for **Car**. Each block shows a user (Cartoon, Winter, B/W, and Sketch) with No-Guidance-DM vs. MMD Guidance.

| User | Guidance | FD ↓ | KD ↓ | RRKE ↓ | Density ($\times 10^2$) ↑ | Coverage ($\times 10^2$) ↑ |
|------|----------|------|------|--------|---------|----------|
| B/W | No-Guidance-DM | $1487.97 \pm 47.90$ | $9.07 \pm 0.60$ | $1.74 \pm 0.05$ | $59.41 \pm 4.56$ | $50.02 \pm 3.42$ |
| | MMD (Ours) | $1242.00 \pm 46.46$ | $6.94 \pm 0.56$ | $1.62 \pm 0.04$ | $55.76 \pm 3.98$ | $55.62 \pm 2.27$ |
| Winter | No-Guidance-DM | $1394.50 \pm 33.74$ | $8.41 \pm 0.39$ | $1.78 \pm 0.03$ | $11.04 \pm 2.74$ | $11.82 \pm 2.01$ |
| | MMD (Ours) | $1127.11 \pm 28.07$ | $6.82 \pm 0.37$ | $1.62 \pm 0.03$ | $15.92 \pm 1.52$ | $20.88 \pm 1.30$ |
| Cartoon | No-Guidance-DM | $2222.78 \pm 13.54$ | $16.80 \pm 0.15$ | $3.35 \pm 0.05$ | $1.72 \pm 1.09$ | $1.36 \pm 0.56$ |
| | MMD (Ours) | $1826.11 \pm 14.44$ | $14.00 \pm 0.18$ | $2.74 \pm 0.04$ | $2.75 \pm 1.15$ | $4.06 \pm 0.86$ |
| Sketch | No-Guidance-DM | $1268.91 \pm 23.54$ | $8.77 \pm 0.25$ | $1.75 \pm 0.03$ | $4.88 \pm 1.30$ | $5.80 \pm 0.861$ |
| | MMD (Ours) | $1207.64 \pm 26.41$ | $8.29 \pm 0.24$ | $1.71 \pm 0.03$ | $7.72 \pm 2.12$ | $9.10 \pm 1.42$ |

polynomial $d=3$ is marginally better on metrics than the RBF choices, but the gap is very small. Moreover, across all settings, performance improves as $\alpha$ increases from 0 up to a small range and then starts to drop. However, the overall results shows that the MMD Guidance is not highly dependent on the choice of guidance parameters or kernel.

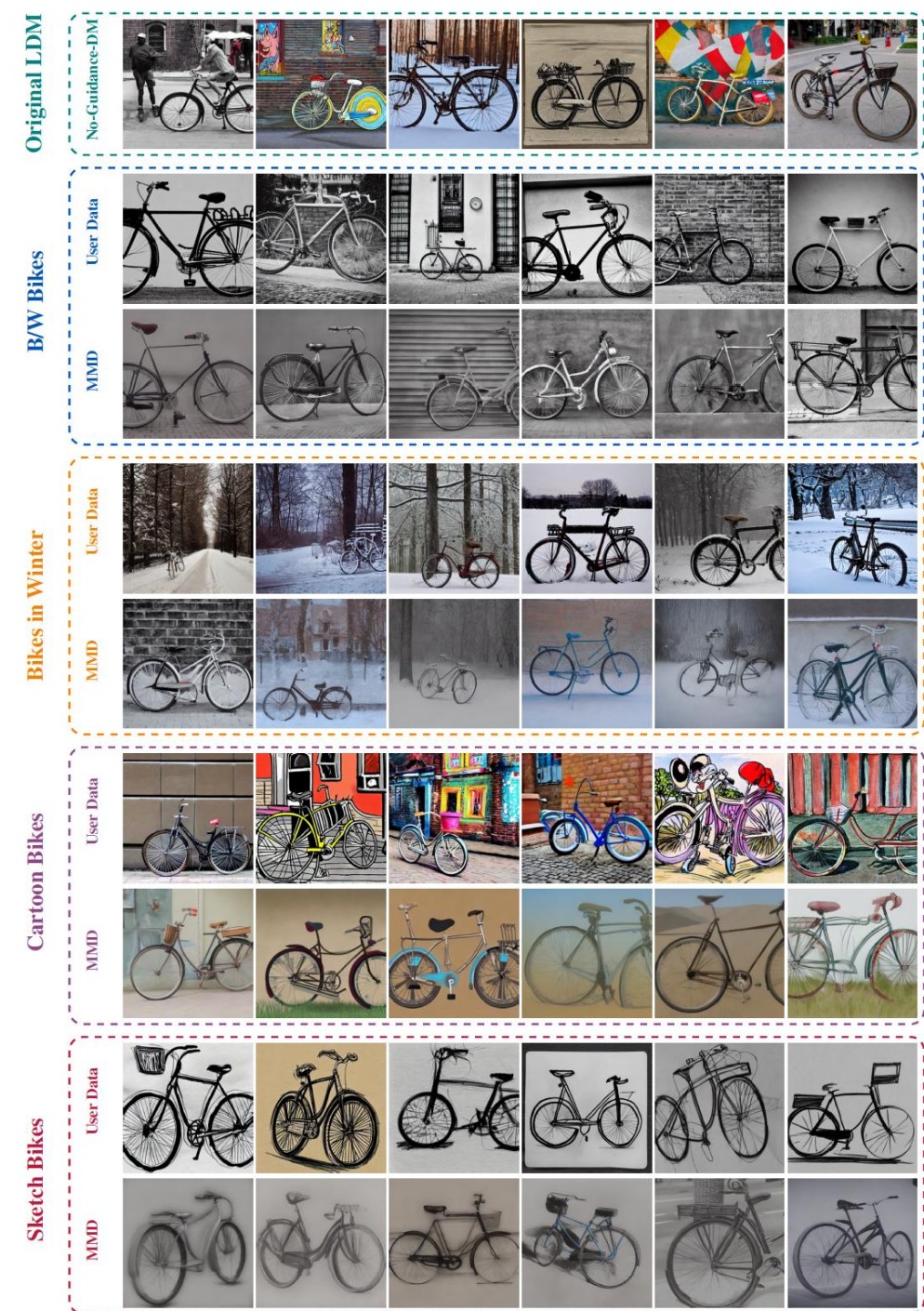

Figure 12: MMD Guidance on synthetic **Bikes** data using SD v.14.

**Ambient space MMD Guidance.** The previous experiments apply MMD Guidance in the latent space, we additionally applied MMD Guidance on ambient space using features, extracted from the Inception v3 (Szegedy et al., 2016) and computing MMD in that feature space. For these experiments we used AFHQ datasets.

**Experiment settings on AFHQ.** We evaluated the metrics listed in Table 16. We conducted experiments on three users derived from AFHQ classes, each with 500 images of dogs, cats, and wild. The user images were passed through the Inception v3 to construct the features used for the MMD Guidance on ambient space. Figures 18 depict extra qualitative results for the MMD Guidance for

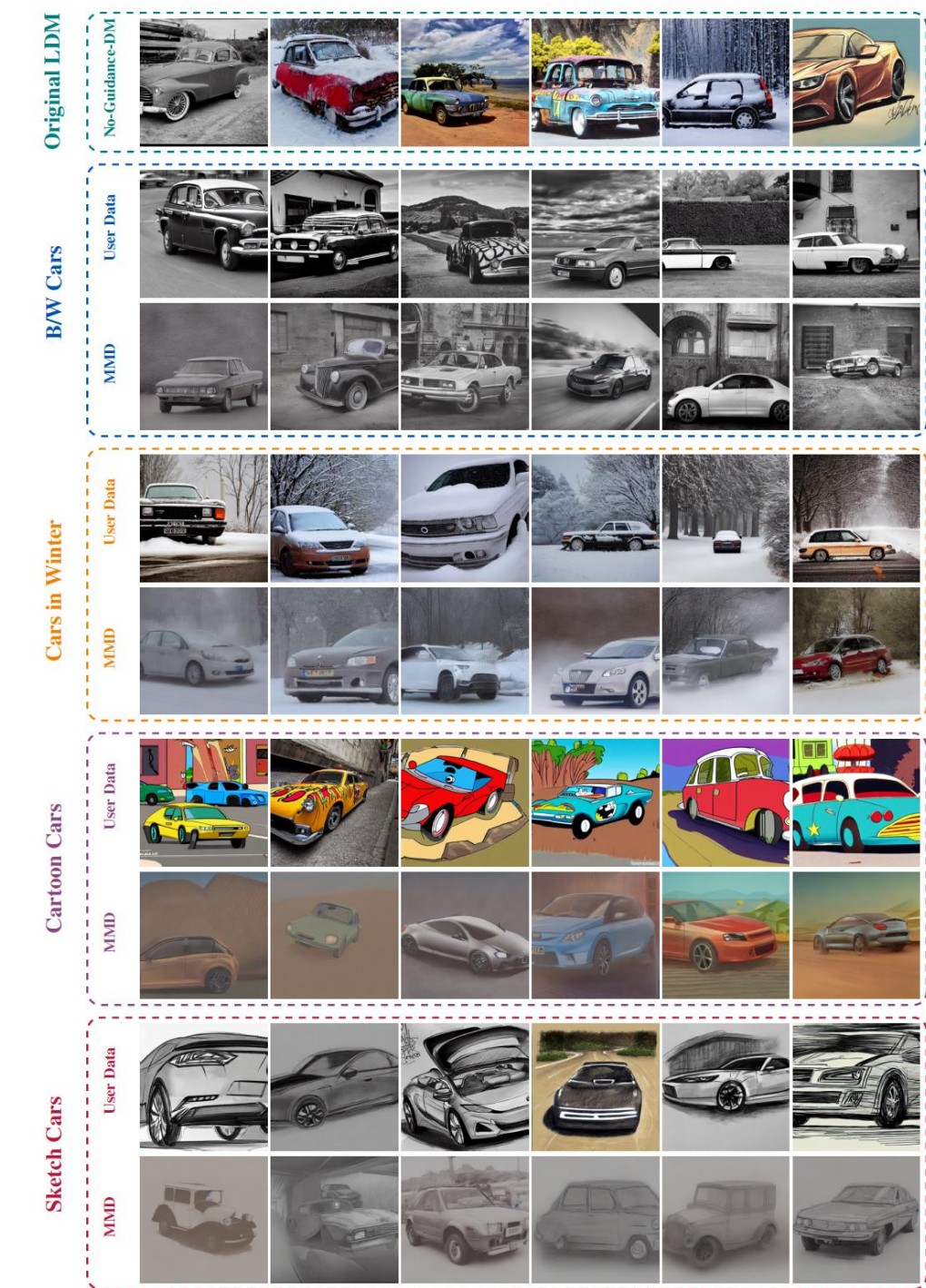

Figure 13: MMD Guidance on synthetic **Cars** data using SD v.14.

these users. For AFHQ experiments, we generated samples using weights from Daras et al. (2023). The MMD Guidance scale is in order of $10^{-1}$, and all metrics are computed on 500 generated samples per method. Table 16 compares the evaluation metrics with the No-Guidance-DM baseline to assess the effectiveness of our method.

**Effect of the number of reference samples on the MMD Guidance.** To assess how the number of available reference sets influences the effectiveness of MMD, we evaluated our method under varying sample sizes for FFHQ and GMM datasets.

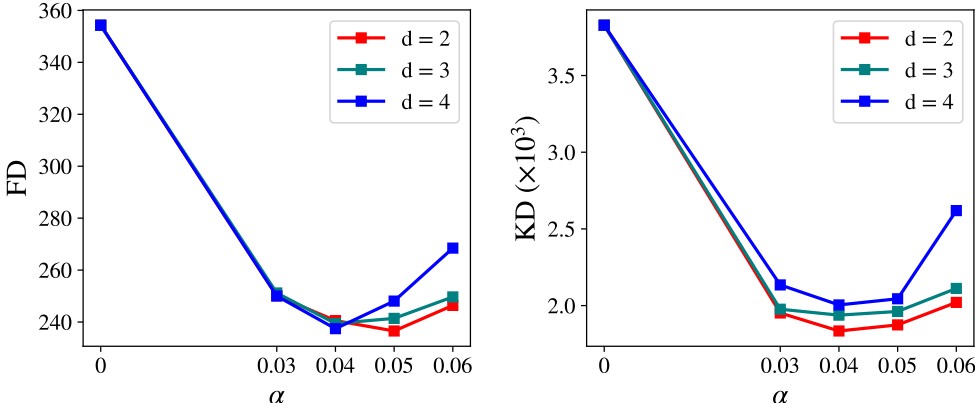

Figure 14: Effect of MMD Guidance scale $\alpha$ with polynomial kernels of degree $d \in \{2, 3, 4\}$ on FD and KD metrics.

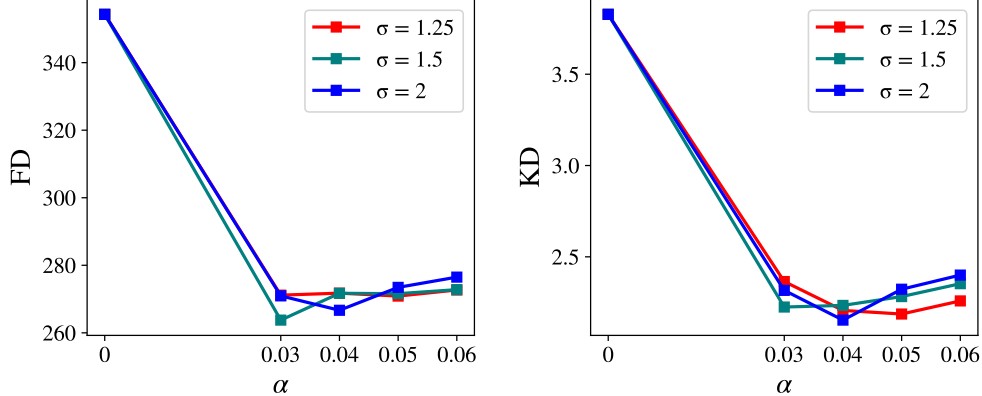

Figure 15: Effect of MMD Guidance scale $\alpha$ with RBF kernels of $\sigma \in \{1.25, 1.5, 2\}$ on FD and KD metrics.

Table 10: Evaluation metrics as a function of the number of reference samples for the Sunglasses user of FFHQ dataset.

| # of ref. samples | Guidance | FD $\downarrow$ | KD $\downarrow$ | RRKE $\downarrow$ |
|---|---|---|---|---|
| $n = 0$ | No-Guidance-DM | 1004.71 | 8.21 | 1.73 |
| $n = 10$ | MMD | 886.41 | 5.67 | 1.55 |
| $n = 50$ | MMD | 736.69 | 3.78 | 1.44 |
| $n = 150$ | MMD | 719.37 | 3.59 | 1.41 |
| $n = 250$ | MMD | 710.48 | 3.33 | 1.41 |
| $n = 400$ | MMD | 696.07 | 3.29 | 1.38 |
| $n = 500$ | MMD | 692.87 | 3.25 | 1.39 |

FFHQ setting: We subsample the sunglasses reference set to $10, 50, 150, 250, 400, 500$ examples and compute all evaluation metrics. As expected, table 10 depicts that the performance improves significantly in compare to no guidance after only 50 reference samples and it converges shortly after.

GMM setting: We study the effect of the number of reference samples using a synthetic GMM dataset with both 8, and 25 components. The reference set is drawn from 3 randomly selected components and further subsampled to obtain reference sets of sizes $5, 25, 50, 100, 150$. We compute all evaluation metrics for each setting. As shown in Table 11, the performance converges rapidly, with little to no improvement beyond 50 reference samples.

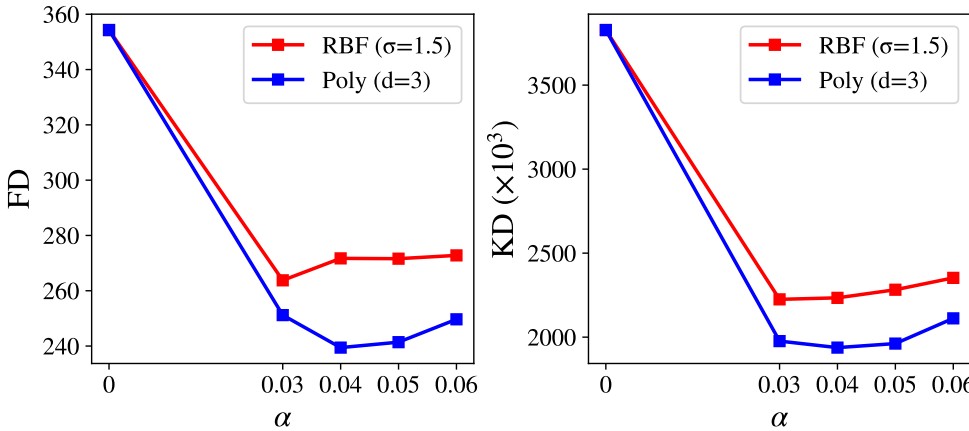

Figure 16: Comparison of RBF ($\sigma = 1.5$) vs. Polynomial (degree $d = 3$) under varying MMD Guidance scale $\alpha$ on FD and KD metrics.

Table 11: Evaluation metrics as a function of the number of reference samples for 8- and 25-component Gaussian mixtures.

| # Ref. Samples | Guidance | 8 Gaussian Component | | | 25 Gaussian Component | | |
|---|---|---|---|---|---|---|---|
| | | FD ↓ | KD($\times 10^3$)↓ | RRKE ↓ | FD ↓ | KD($\times 10^3$)↓ | RRKE ↓ |
| $n = 0$ | No-Guidance-DM | 7.04 | 142.42 | 2.31 | 77.70 | 330.91 | 2.073 |
| $n = 5$ | MMD | 4.62 | 65.17 | 1.45 | 29.63 | 79.72 | 0.81 |
| $n = 25$ | MMD | 0.49 | 29.72 | 1.33 | 4.93 | 22.66 | 0.33 |
| $n = 50$ | MMD | 0.38 | 24.51 | 1.30 | 3.52 | 9.49 | 0.23 |
| $n = 100$ | MMD | 0.34 | 23.33 | 1.22 | 3.69 | 11.25 | 0.25 |
| $n = 150$ | MMD | 0.37 | 17.02 | 1.29 | 3.56 | 9.52 | 0.23 |

Table 12: Evaluation metrics as a function of the number of reference samples.

| # of ref. samples | FD ↓ | KD ↓ | RRKE ↓ |
|---|---|---|---|
| $n = 10$ | 1715.86 | 1.5944 | 1.98 |
| $n = 50$ | 1709.59 | 1.5498 | 1.92 |
| $n = 100$ | 1693.35 | 1.5082 | 1.81 |
| $n = 150$ | 1681.33 | 1.4887 | 1.79 |
| $n = 200$ | 1679.69 | 1.4534 | 1.76 |

Table 13: Comparison of SDXL variants with and without MMD guidance.

| Model | FD ↓ | KD ↓ | RRKE ↓ |
|---|---|---|---|
| SDXL (No-Guidance-DM) | 1734.25 | 1.5536 | 1.84 |
| SDXL + MMD Guidance (clean reference) | 1678.24 | 1.4429 | 1.76 |
| SDXL + MMD Guidance (separate steps) | 1648.95 | 1.4405 | 1.62 |

**Applying MMD guidance and denoising process separately.** We conducted an experiment in which MMD guidance was applied separately from the denoising process. We performed five MMD-guidance steps every ten diffusion timesteps. As shown in Table 13, this modification yielded improved MMD-guidance performance compared with the original simultaneous-guidance approach.

**Computing MMD guidance with noisy reference sample.** Prompt-aware results: We compared computing MMD guidance with clean reference samples and with the latent vector of the reference sample at time step $t$. Suggested by the results in Table 14, we observe that using the latent vector of the noisy reference sample at time-step $t$, achieves comparable results to using clean references.

**MMD Guidance for adapting highly different domains.** To test MMD guidance in highly different domains, following Somepalli et al. (2024), we used two distinct painting styles, Klimt and

Table 14: Comparison of SDXL variants with and without MMD guidance.

| Model | FD ↓ | KD ↓ | RRKE ↓ |
|---|---|---|---|
| SDXL (No-Guidance-DM) | 1734.25 | 1.5536 | 1.84 |
| SDXL + MMD Guidance (noisy reference at timestep $t$) | 1725.56 | 1.5487 | 1.79 |
| SDXL + MMD Guidance (clean reference) | 1678.24 | 1.4429 | 1.76 |

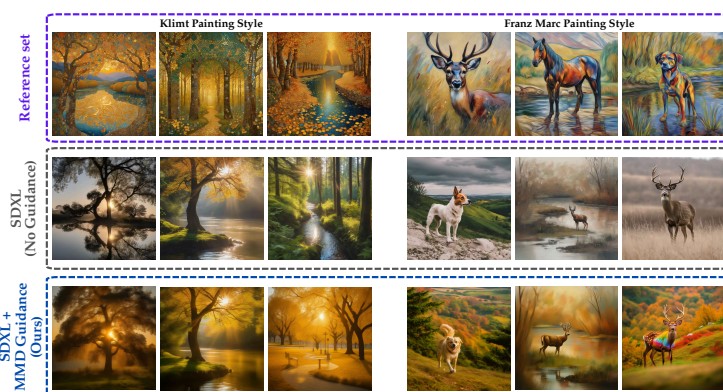

Figure 17: Caption

Franz Marc, as the reference set and applied MMD guidance to adapt the distribution of samples generated by SDXL. To quantitatively measure the domain gap between SDXL outputs and the reference set, we computed Recall and Coverage, obtaining values of 0.34 and 0.12, respectively. In the following Table, we report results with/without MMD guidance.

**Effect of Reference Mixture Weights on Mode Proportions.** To address whether guidance can adjust mixture proportions, an additional experiment was conducted on the synthetic mixture-of-Gaussians setup. A diffusion model was first trained on data where all Gaussian components had equal mixture weights, yielding a uniform distribution over modes. As a baseline, samples generated without any guidance (second panel) reproduce this uniform mixture and do not reflect any target reweighting.

To simulate a non-uniform target distribution, reference samples were drawn from a mixture whose component weights were sampled from a Dirichlet distribution with parameter (1,10) over the depicted components, thereby concentrating mass on a subset of modes. As depicted in Figure 3 after applying MMD-Guidance at sampling time, the guided diffusion model successfully shifted its outputs to match these reweighted proportions: the generated samples predominantly occupy the emphasized component, and their relative frequencies closely track those in the reference set. This demonstrates that the proposed guidance mechanism can meaningfully steer mode proportions using only a small, reweighted reference sample set. Table 15 provides numerical proofs for this observation. After generation, each sample is assigned to the closest GMM component using a $k$-nearest-neighbor classifier.

**Additional qualitative results for Prompt-based diffusion models.** We conducted experiments on Pixar animated cars and Cowboy animated horses using Pixart-$\Sigma$ diffusion models, similar to the settings shown in Figure 1 as illustrated in Figure 19. This comparison evaluates image generation via a text-conditioned latent diffusion model (LDM) with and without guidance. The LDM (Pixart-$\Sigma$) employing our proposed MMD guidance, based on 100 reference samples of "car" and "horse" images, effectively captures the visual format of the target distribution. In contrast, the unguided LDM outputs exhibit stylistic differences from the target model.

# D STATEMENT ON LLM USAGE

LLMs were used only for proofreading and polishing the writing. All research ideas, theoretical results, algorithms, experiments, and analyses were executed by the authors without the use of LLMs.

Table 15: Effect of mode proportions in MMD guidance on the number of samples in each GMM component.

| Guidance | 1 | 2 | 3 | 4 | 5 | 6 | 7 | 8 |
|---|---|---|---|---|---|---|---|---|
| Reference sample | 90% | 10% | 0% | 0% | 0% | 0% | 0% | 0% |
| No-Guidance-DM | 12% | 12% | 11% | 18% | 12% | 10% | 13% | 12% |
| MMD | 88% | 12% | 0% | 0% | 0% | 0% | 0% | 0% |

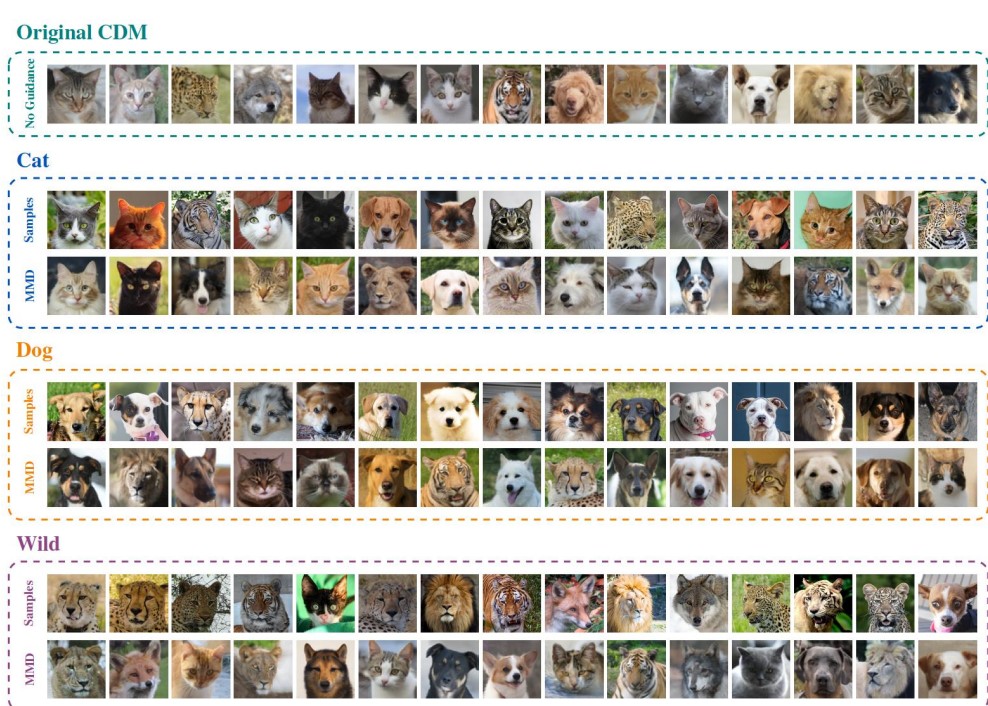

Figure 18: MMD Guidance on AFHQ dataset.

Table 16: Evaluation metrics for AFHQ with No-Guidance-DM vs. MMD Guidance.

| User | Guidance | FD ↓ | KD ↓ | RRKE ↓ | Density ($\times 10^2$) ↑ | Coverage ($\times 10^2$) ↑ |
|---|---|---|---|---|---|---|
| Cat | No-Guidance-DM | $1154.87 \pm 231.71$ | $3.16 \pm 1.17$ | $1.37 \pm 0.10$ | $44.70 \pm 20.08$ | $38.16 \pm 2.99$ |
| | MMD (Ours) | $1030.14 \pm 208.31$ | $2.85 \pm 1.02$ | $1.33 \pm 0.08$ | $42.81 \pm 18.25$ | $43.08 \pm 2.37$ |
| Dog | No-Guidance-DM | $1309.05 \pm 271.71$ | $2.55 \pm 0.56$ | $1.40 \pm 0.08$ | $42.21 \pm 10.34$ | $60.96 \pm 7.94$ |
| | MMD (Ours) | $1147.64 \pm 203.07$ | $1.91 \pm 0.36$ | $1.36 \pm 0.06$ | $42.50 \pm 7.47$ | $63.60 \pm 5.38$ |
| Wild | No-Guidance-DM | $1301.12 \pm 176.96$ | $4.69 \pm 1.26$ | $1.39 \pm 0.06$ | $41.87 \pm 16.39$ | $21.64 \pm 6.46$ |
| | MMD (Ours) | $1052.44 \pm 115.79$ | $3.45 \pm 0.90$ | $1.32 \pm 0.04$ | $35.47 \pm 14.14$ | $24.96 \pm 6.21$ |

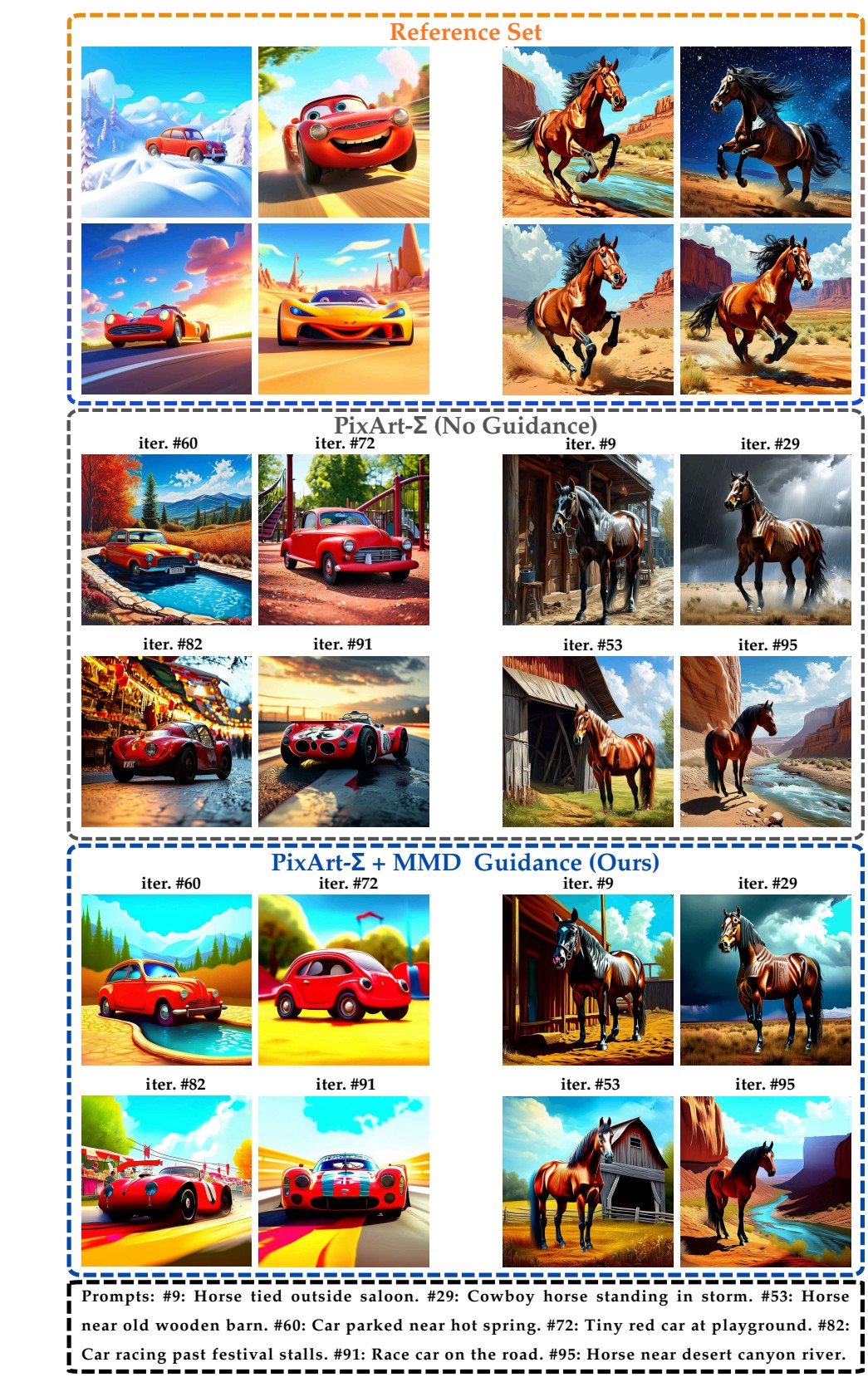

Figure 19: Comparison of Pixart-Σ image generation with and without MMD guidance, showing style differences in unguided LDM outputs from the target distribution of "car" and "horse" images.

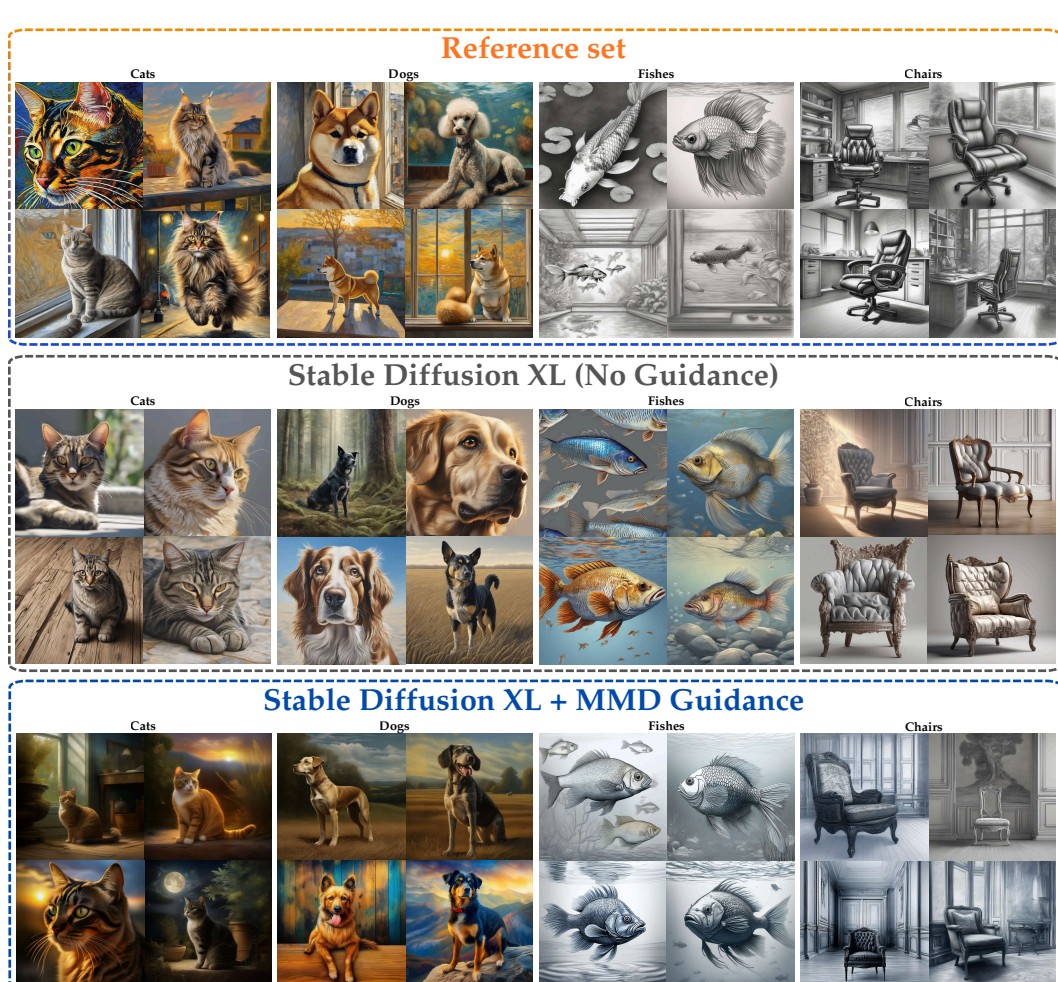

Figure 20: Qualitative comparison of reference set and MMD-guided image generation with SDXL.

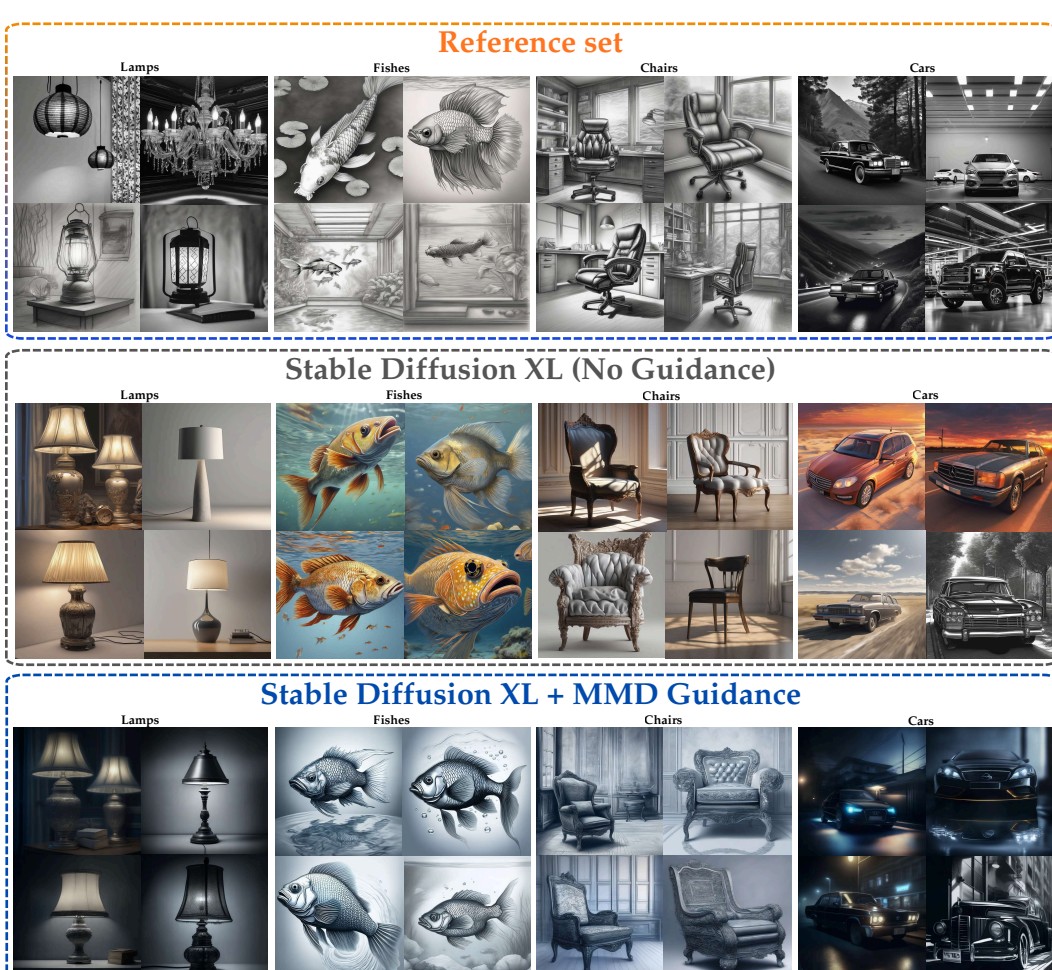

Figure 21: Qualitative comparison of reference set and MMD-guided image generation with SDXL.

