# OpenReview forum: "Training-Free Distribution Adaptation for Diffusion Models via Maximum Mean Discrepancy Guidance"
_ICLR.cc/2026/Conference — Submitted to ICLR 2026_

### Official Review · Reviewer_zL3j · 2025-10-25

**Soundness:** 4
**Presentation:** 4
**Contribution:** 3
**Rating:** 8
**Confidence:** 3

**Summary:**

The paper proposes a method to guide a pretrained (possibly conditional) diffusion model using reference samples. Guidance is introduced by subtracting the gradient of a maximum mean discrepancy term (computed between model samples and reference samples) during the sampling process. The authors motivate the use of MMD, providing both theoretical justification and intuitive insight into the effect of this gradient correction. The approach is evaluated on synthetic Gaussian mixture experiments as well as on image generation tasks, in both unconditional and prompt-aware settings, showing clear improvements in sample alignment with the reference data.

**Strengths:**

* The paper is clearly written and easy to follow.
* The proposed idea is simple, elegant, well-motivated, and computationally efficient.
* The experiments convincingly demonstrate the method’s effectiveness in well-controlled synthetic settings using Gaussian mixtures.

**Weaknesses:**

* The paper provides no formal proof that the kernel in Eq. (5) converges to, or accurately approximates, the kernel corresponding to the denoising of the reference distribution.

**Questions:**

- I have questions on the MMD considered itself. Why not denoting the reference distribution $\hat{Q} = N^{-1} \sum_{j = 1}^N \delta_{\tilde{z}^{(j)}}$ where the dataset of the latent reference samples is $(\tilde{z}^{(j)})_{j=1}^N$ ? It seems more consistent with the notation on $P_t$. Moreover, why computing the MMD between noisy generated samples and clean reference samples (both in latent space) ? Wouldn't it be more correct to compute the distance between the noisy generated samples and the noisy reference samples ? Or maybe between the samples obtained by the denoiser at sampling step $t$ and the clean reference samples ?
- In section 4.2, you show that what is actually doing the job is the action $z \leftarrow z - \lambda_t \nabla \widehat{\operatorname{MMD}^2}$ but thats not what Eq. 5 implements. Why not actually **composing** the classic sampling steps with MMD gradient descent steps ? It would also raise the question of doing multiple MMD GD steps after a single sampling step.
- Have you investigated the impact of the number of reference samples ?
- Coming back to the synthetic MoG experiment, it would be interesting to see if, given a diffusion model trained on a MoG with equal weights on each mode, you can use reference samples with different weighting of each mode to change the proportions. It seems like a difficult task.

---

> ### Author Response · Authors · 2025-11-26
>
> We sincerely thank Reviewer zL3j for the thoughtful feedback and constructive suggestions on our work. We are pleased that the reviewer finds our work “clearly written and easy to follow” and our proposed idea to be “elegant, well-motivated, and computationally efficient”. Below, we provide our responses to the comments and questions in the review.
>
> **1- W1: Theoretical convergence analysis of the kernel in Eq. (5)**
>
> We would like to clarify that we have decomposed the Gradient of the square of MMD distance in Eq. (5) into the following two components: 1) the intra-batch term including only the generated data  and 2) the cross term with both generated and reference samples in Eq. (6). Noting that the intra-batch term is only between the generated samples and is independent of the reference samples, we focus our convergence analysis for the second component, which is the only term affected by the reference data size.
>
> In Theorem 1, we show that the empirical cross-term gradient based on $N_r$ reference samples concentrates around its population value, and in Theorem 2, we establish a uniform concentration bound for these gradients. Also, as we argued in the response to Item 3, Reviewer eNzo, the application of Theorem 1’s bound over $T$ rounds of sample generation will only add an $O(\sqrt{\log(T)})$ multiplicative term to the concentration bound in Theorems 1 and Corollary 1, which would improve the concentration bounds over that of Theorem 2, assuming a reasonable number of rounds of generating data $T$. We will clarify this point in the revision.
>
> **2- Computing MMD guidance with noisy reference samples**
>
> We thank the reviewer for raising this relevant point. We appreciate the suggestion that the guidance can be towards the noisy reference distributions in the middle reverse diffusion steps. We have tested the idea and the numerical results were comparable to the results of noise-free reference samples in the guidance process.  Specifically, we compared the MMD guidance computed with clean reference samples vs. the noisy reference samples (noise matched with time step t). We observe that using noisy reference samples at time-step $t$, achieves comparable results to using clean references.
>
> | Model    | FD $\downarrow$ | KD $\downarrow$ | RRKE $\downarrow$ |
> |-------|:----:|:---------:|:---------:|
> | SDXL  (No Guidance)    | 1953.75         | 3.57          | 1.93             |
> | SDXL + MMD Guidance (noisy reference at timestep t) | 1725.56       | 2.6492        | 1.83     |
> | SDXL + MMD Guidance (clean reference) | 1674.45         | 2.4945       | 1.79          |
>
> **3- Applying MMD guidance and denoising process using alternating optimization**
>
> We thank the reviewer for the great suggestion, which resembles the idea of alternating training of the generator and discriminator players in GANs. Following the suggestion, we tested the idea of alternating the reverse diffusion denoising (sampler) step and MMD guidance steps, where instead of *simultaneous* application of sampler and MMD guidance, we perform them *alternatingly*. We note that our current implementation of MMD guidance follows the standard guidance schemes in diffusion models, where the gradient of the potential function (MMD distance-squared in our case) is simultaneously considered in the guidance process.
>
> As explained, we followed the reviewer’s suggestion and conducted an experiment in which MMD guidance was applied alternatingly with the denoising process. We performed 5 MMD-guidance steps following every 10 reverse diffusion timesteps. This modification could moderately improve MMD-guidance performance in SD-XL prompt-aware generation compared to our original simultaneous-guidance approach. We will include this discussion in the revision
>
> | Model     | FD $\downarrow$ | KD $\downarrow$ | RRKE $\downarrow$ |
> |--|--|----|---|
> | SDXL  (No Guidance)     | 1953.75  | 3.57  | 1.93   |
> | SDXL + MMD Guidance (simultaneous steps) | 1674.45   | 2.49   | 1.79    |
> | SDXL + MMD Guidance (alternating steps) | 1648.95         | 2.47  | 1.62    |

---

> ### Author Response · Authors · 2025-11-26
> **Official Comment by Authors (continued)**
>
> **4- Impact of the number of reference samples**
>
> Following the question of the reviewer, we have numerically tested the performance of MMD guidance and report the results in the following table for the experiments of Figures 2, 3 in the submission. As can be seen, the performance changes significantly when the number of reference samples varies from 0 (no MMD guidance) to 100 and then the performance looks mildly improving with more reference data. We will include the table in the revised draft.
>
> **Table 1:** The following are the numerical results for the FFHQ case:
>
> |User|#Reference Samples|FD↓|KD↓|RRKE↓|
> |:----:|:------:|:----:|:----:|:----:|
> |Sunglasses|0 (No MMD Guidance)|1004.71|8.21|1.73|
> |Sunglasses|10|886.41|5.67|1.55|
> |Sunglasses|50|736.69|3.78|1.44|
> |Sunglasses|150|719.37|3.59|1.41|
> |Sunglasses|250|710.48|3.33|1.41|
> |Sunglasses|400|696.07|3.29|1.38|
> |Sunglasses|500|692.87|3.25|1.39|
>
> **Table 2:** Here are the numerical results for the prompt-aware SD-XL case:
>
> | # of reference samples | Guidance | FD $\downarrow$ | KD $\downarrow$ | RRKE $\downarrow$ |
> |:-----:|:---:|:----:|:------:|:------:|
> | n = 0         | No Guidance | 1953.75  |    3.57         | 1.93              |
> | n = 10        |    MMD      | 1805.92  | 2.7657          | 1.89              |
> | n = 50        |    MMD      | 1731.27 | 2.6882          |  1.83             |
> | n = 100       |    MMD      | 1691.42   | 2.5682          |  1.81             |
> | n = 150       |    MMD      | 1686.61  | 2.5301          |  1.80             |
> | n = 200       |    MMD      | 1674.45         | 2.4945          |  1.79             |
>
> **Table 4:** The following are the numerical results for the synthetic case of Gaussian mixtures:
>
> | | | |8 Gaussian Component|||25 Gaussian Component||
> |-|-|:-:|:-:|:-:|:-:|:-:|:-:|
> |**#References**|**Guidance**|**FD $\downarrow$** |**KD(1000) $\downarrow$**| **RRKE $\downarrow$** | **FD $\downarrow$**| **KD(1000) $\downarrow$** | **RRKE $\downarrow$** |
> |0|No Guidance|7.04|142.42|2.31|77.70|330.91|2.073|
> |5|MMD|4.62|65.17|1.45|29.63|79.72|0.81|
> |25|MMD|0.49|29.72|1.33|4.93|22.66|0.33|
> |50|MMD|0.38|24.51|1.30|3.52|9.49|0.23|
> |100|MMD|0.34|23.33|1.22|3.69|11.25|0.25|
> |150|MMD|0.37|17.02|1.29|3.56|9.52|0.23|
>
> **5- Changing Mode Proportions in MoG**
>
> We thank the reviewer for the insightful question. Indeed, our experiments of MMD guidance suggest that the method is capable of addressing the mismatch of the weights of Gaussian mixture components. We highlight that the MMD-distance-squared with the RBF kernel could play the role of a proxy for the discrepancy between the weights of the original and reference GMM distributions.
>
> To numerically verify this, we conducted an experiment where the training samples were uniformly distributed across the 8 Gaussian components (8-GMM case). The second row of the following table shows the generated samples without any guidance. The reference data distribution was simulated by sampling mixture weights from a Dirichlet distribution with parameters (1, 10) between two random components. The following table shows the percentage of samples in each GMM component after applying MMD. Samples are assigned to components using K-nearest neighbours. This demonstrates that MMD not only guides samples toward the desired components but also matches the component proportions of the reference samples.
>
> |     | 1   | 2   | 3   | 4   | 5   | 6   | 7   | 8   | FD   $\downarrow$    | KD ×1000  $\downarrow$  | RRKE  $\downarrow$    |
> |:--|:--|:--|:--|:--|:--|:--|:--|:--|:--:|:--:|:--:|
> | Reference sample | 90% | 10% | 0%  | 0%  | 0%  | 0%  | 0%  | 0%  | - | -   | -  |
> | No guide         | 12% | 12% | 11% | 18% | 12% | 10% | 13% | 12% | 4.22 | 29.64 | 1.94  |
> | MMD              | 87.9% | 12.1% | 0%  | 0%   | 0%   | 0%   | 0%   | 0%   | 0.13 | 2.38   | 0.40 |

---

> ### Comment · Reviewer_zL3j · 2025-11-28
> **Answer to the rebuttal**
>
> I thank the authors for their detailed and thoughtful rebuttal. I would like to clarify one point where I believe my original question may have been misunderstood. My concern regarding the theoretical guarantees of Eq. (5) was not about the concentration properties of the empirical MMD gradient, but rather about whether the denoising kernel induced by Eq. (5) is provably close to the true denoising kernel associated with the reference distribution. In other words, can one formally justify that the modified reverse dynamics approximate the reverse dynamics that would arise if the diffusion model had been trained directly on the reference distribution? This is the aspect for which I still do not see a theoretical argument.
>
> Regarding the empirical behavior of the MMD guidance, I appreciate the experiments with noisy reference samples. I admit I am surprised that using noisy versus clean reference yields nearly identical results, since these correspond to substantially different target distributions. This makes the finding even more interesting.
>
> Thank you as well for testing the alternating optimization scheme and for providing the ablation on the number of reference samples; both sets of experiments are informative. I especially enjoyed the mode-proportion experiment in the GMM setting,it compellingly demonstrates the ability of MMD guidance to modulate mixture weights, which is a nontrivial property.
>
> Overall, the rebuttal satisfactorily addresses my main concerns. I have carefully read the responses to all reviewers, and I will keep both my score and my confidence level, acknowledging that I am somewhat less specialized in diffusion-model guidance than some of the other reviewers.

---

> ### Author Response · Authors · 2025-11-28
>
> We sincerely thank Reviewer zL3j for the positive feedback on our response. We are glad that the response could address the reviewer’s main comments and questions.
>
> Also, we thank the reviewer for the helpful clarification about the question on Eq. (5). We would like to clarify that our theoretical analysis does not claim that the MMD-guided reverse kernel coincides with the ideal reverse kernel of a diffusion model trained directly on the reference distribution. Instead, our theoretical guarantee concerns a different, more structural property: *When the distribution of generated samples matches the reference distribution, the population MMD guidance field becomes identically zero*.
>
> The MMD guidance drift for a particle at position $x$ takes the form where $Z$ follows distrbution $q$ and $p$ in the following expectations:
>
> $g(x)\propto \mathbb{E}_{q}[\nabla_x k(x,Z)]-{\mathbb{E}}_p[\nabla_x k(x,Z)]$
>
> Therefore, whenever $q = p$, the two expectations coincide and the drift vanishes pointwise: $g(x) = 0$ for all $x$. Consequently, the true reference distribution $P_{\text{ref}}$ is a *stable fixed point* of the deterministic MMD-guided reverse flow. This shows that MMD guidance does not bias the reverse process away from the correct target distribution in the population limit.
>
> However, this fixed-point property is different from the stronger statement the reviewer is asking about, namely, that the *entire* reverse-time denoising kernel induced by MMD guidance is close to the kernel that would arise from training a diffusion model directly on $P_{\mathrm{ref}}$. Establishing such a result would require a detailed characterization of the reverse dynamics of DDPM-type samplers under specific kernel structures, which lies beyond the current theoretical analysis. We believe this is an interesting direction for future theoretical work on MMD guidance and diffusion-model dynamics.

---

### Official Review · Reviewer_7KWN · 2025-10-29

**Soundness:** 3
**Presentation:** 3
**Contribution:** 3
**Rating:** 4
**Confidence:** 4

**Summary:**

The authors introduced the MMD Guidance method to help diffusion models create images that match a user's style without needing retraining. This method adds a new component during the sampling step that changes the direction of image generation.
This approach uses a new gradient component (MMD^2). It shifts the generation process toward the user's data by comparing the current distribution of model samples with the user's distribution. This gradient also improves the diversity of the generated samples.
The paper looks at the results and usefulness of this method in two cases: unconditional generation and prompt-aware diffusion. It provides detailed results and comparisons using synthetic data (GMM), face datasets (FFHQ and CelebA-HQ), and models like SDXL and PixArt.

**Strengths:**

a) No retraining is required
b) It is easy to implement and understand
c) It has low computational costs
d) It is based on a strong theoretical foundation
e) The method is clearly described
f) It produces very good visual and quantitative results (FD, KD, Coverage)
g) The experiments are reliable, with results averaged from five random tests

**Weaknesses:**

a) It cannot learn new styles that the diffusion model doesn't know
b) Hyperparameter values ​​(guidance strength, kernel bandwidth) are selected empirically, which can hinder the correct generation of images
c) The experiments mainly focus on simple prompts, like "person with sunglasses," so its effectiveness for more complex prompts is not clear.

**Questions:**

1. How does the computational cost of MMD Guidance scale with the number of reference samples?
2. How sensitive is the method to the choice of guidance strength and kernel bandwidth?
3. Could adaptive or data-driven tuning of these hyperparameters improve robustness?
4. Can MMD Guidance handle complex or compositional prompts beyond simple cases?
5. Does the method provide convergence guarantees for distribution alignment during sampling?
6. Have the authors explored alternative kernels that may better capture perceptual similarity?
7. How does the proposed approach compare to other training-free guidance methods like:
- Classifier-free Guidance with Adaptive Scaling
- Learn to Guide Your Diffusion Model
- CFG++: Manifold-constrained Classifier Free Guidance for Diffusion Models
8. Could this method be extended to other modalities such as audio, video, or 3D data?

---

> ### Author Response · Authors · 2025-11-26
>
> We sincerely thank Reviewer 7KWN for the thoughtful feedback and constructive suggestions on our work. We are pleased that the reviewer finds the method to be “based on a strong theoretical foundation” and producing “very good visual and quantitative results”.  In the following, we give our responses to the comments and questions in the review.
>
> **1- The growth of computational costs of MMD guidance with the number of reference samples**
>
> We would like to clarify that the computational costs of MMD guidance grow only linearly $\mathcal{O}(N_r)$ with the number of reference samples $N_r$. This property follows the expansion of the MMD objective function, where the number of terms with non-zero $z_t$-based gradient is $N_r + B$ ($B$ is the generated data size). This has been shown in Equation (6) for prompt-free and Equation (11) for prompt-aware cases. As a result, the computational cost remains affordable even with very large $N_r$’s.
>
> **2- Questions 2,3,6 on the ablation analysis**
>
> We appreciate the reviewer’s questions regarding the sensitivity of our method to the guidance strength, kernel bandwidth, and kernel choice. We would like to kindly refer the reviewer to the ablation studies in Appendix Figures 13, 14, and 15 which address the impact of these parameters on FD and KD metrics. To further address the reviewer’s question, we will move some of the ablation results from the Appendix to the main text.
>
> **3- MMD guidance performance on complex prompts**
>
> We would like to clarify that, in the prompt-aware MMD guidance experiments, the prompts were generated using GPT-4o, resulting in complex prompt structures. These prompts are included in Figures 17 and 18 of the Appendix. In the following, we have provided randomly selected prompts in our experiments to highlight the complexity of the prompts in our experiments:
>
> - Cats: “A cat watching stars in the night sky from a rooftop balcony”, “A cat wandering through tall wildflowers at sunset”, “A cat wandering through tall bamboo in a misty forest”
> - Horses: “Horse standing outside a busy train station at dusk”, “Horse galloping past a canyon river under stormy clouds”, “Cowboy horse standing near a campfire under starry skies”, “Horse galloping near desert cliffs at sunrise”
> - Dogs: “A dog wandering through a crowded festival street at night”, “A dog sitting in front of a shrine lantern in the rain”, “A dog sitting under cherry blossoms in a quiet park”
> - Cars: “Car parked beside a festival gate on a rainy evening”, “Car drifting through neon city streets at midnight”, “A car crossing a wooden bridge over a foggy river”

---

> ### Author Response · Authors · 2025-11-26
> **Official Comment by Authors (continued)**
>
> **4- Comparison with recent works on training-free guidance**
>
> We thank the reviewer for bringing these recent works to our attention. We will cite and discuss these papers in the revision. In the following, we compare and contrast the methods in the papers with our proposed MMD guidance. This discussion will appear in the revision.
>
> Malarz et al. [1] introduce β-CFG, which controls the impact of guidance during the denoising process. Similarly, CFG++ [2] aims to mitigate the off-manifold behavior associated with standard CFG. On the other hand, this work uses MMD guidance to adopt the distribution of generated samples to the reference set. Integrating such techniques to control the strength of MMD Guidance could be an interesting direction for future research.
>
> In the concurrent work, Zhang et al. [3] replace the static guidance weight $w$ with a timestep-dependent learnable function. They train a neural network to predict this function using MMD as the optimization objective. In contrast, our method applies MMD directly within the denoising process to guide samples toward the reference distribution without training any additional parameters or models.
>
> **5-(Q8)Extension of MMD guidance to other modalities**
>
>
> We highlight that MMD guidance can be applied to any latent diffusion model (LDM). In our experiments, we mainly focused on the standard and state-of-the-art image LDMs, as the metrics for evaluating the fidelity and quality of generated image data are standard and well-developed in the literature. However, the MMD distance is a general distance metric for all probability distributions, and in the latent space of an LDM, where the $\ell_2$-norm of a perturbation is supposed to correlate well with the semantic change in the output, the MMD-based distribution matching can similarly be applied. As said, extending the application of MMD guidance to large-scale Audio-LDM [4] and Video-LDM [5] will be an interesting future extension of our numerical study on state-of-the-art image LDMs. We will discuss this point in the revised conclusion.
>
> ---
>
> [1] Malarz et al. “Classifier-free Guidance with Adaptive Scaling”. arXiv 2025
>
> [2] Chung et al. “CFG++: Manifold-constrained Classifier Free Guidance for Diffusion Models”. ICLR 2025
>
> [3] Galashov et al. “Learn to Guide Your Diffusion Model”. arXiv 2025
>
> [4] Liu et al, “AudioLDM: Text-to-Audio Generation with Latent Diffusion Models”, ICML 2023
>
> [5] Blattmann et al, “Align your Latents: High-Resolution Video Synthesis with Latent Diffusion Models”, CVPR 2023

---

### Official Review · Reviewer_5JBa · 2025-10-31

**Soundness:** 2
**Presentation:** 2
**Contribution:** 2
**Rating:** 4
**Confidence:** 5

**Summary:**

This paper proposes MMD Guidance, a training-free adaptation method that directly guides the reverse process of a diffusion model using the gradient of the Maximum Mean Discrepancy (MMD) between generated samples and a reference dataset. The authors argue that MMD provides reliable distributional estimates from limited data, exhibits low variance in practice, and is efficiently differentiable, making it well suited for guidance. The experiments show that MMD Guidance can achieve distributional alignment while preserving sample fidelity.

**Strengths:**

* MMD Guidance is intuitive: it leverages MMD to guide transfer tasks in a training-free manner.
* Theoretical derivations and toy examples are sensible and support the effectiveness of MMD Guidance.
* The experiments show that MMD Guidance outperforms the CG and fine-tuning baselines on the FFHQ benchmarks.

**Weaknesses:**

* An important reference is missing [1], which also presents a guidance-based framework for adapting a pre-trained diffusion model. Reference [1] can be regarded as the classifier-free variant of the DomainCG baseline in Table 1, and is also related to the fine-tuning baseline in Table 5. The authors should carefully discuss [1] and include it as a baseline.
* As a training-free approach, MMD Guidance may struggle to adapt to domains that are quite different from the pre-training domain. Could the authors provide adaptation tasks with a large domain gap so we can fairly assess adaptation capability in boundary scenarios?
* MMD Guidance relies on accurate computation of the gradient of the distance metric. Could the authors discuss in more detail how the size of the reference dataset influences performance, and analyze the computation–precision trade-off as the number of reference samples increases?
* Unreliable guidance signals on noisy data points are a well-known obstacle in CG. How do reference datasets at different noise levels influence the MMD calculation?
* Guidance approaches that rely on explicit gradients often suffer from "gradient hacking." How does MMD Guidance avoid the reverse process exploiting the MMD loss and producing biased samples?

[1] Domain Guidance: A Simple Transfer Approach for a Pre-trained Diffusion Model. ICLR 2025.

**Questions:**

See weakness 2,3,4,5.

---

> ### Author Response · Authors · 2025-11-26
>
> We sincerely thank Reviewer 5JBa for the thoughtful feedback and constructive suggestions on our work. We are glad that the reviewer finds the MMD guidance “intuitive” and our theoretical derivations “sensible and support the effectiveness of MMD Guidance”. In the following, we give our responses to the comments and questions in the review.
>
> **1- W1: Discussion of Domain Guidance baseline**
>
> We thank the reviewer for letting us know about the recent Domain Guidance scheme. As suggested by the reviewer, we have run the experiments in the paper with the Domain Guidance baseline and will add the numerical results to the revised paper. In our experiments, Domain Guidance performed better than the fine-tuning and Classifier-free baselines, as the method can effectively combine the strengths of the two approaches.
>
> Still, in the tasks of updating the fraction of a mode in a mixture distribution, we observed that MMD Guidance can lead to better results, as the method targets the weight mismatch directly in the guidance process. The following table includes the results of our FFHQ experiments (current Table 5 in the main text) with the domain guidance method, which will add to the revised text:
>
>
> |  User  |    Guidance   |     FD ↓   |   KD ↓  |  RRKE ↓   | Density (×100) ↑     |  Coverage (×100) ↑     |
> |:-----:|:-----:|:-------:|:--------:|:------:|:------:|:-----:|
> |     Sunglasses    |      No-Guidance-DM    |     1220.51 ± 131.78    |     8.21 ± 1.79    |     1.73 ± 0.16    |        74.18 ± 23.43      |        30.88 ± 11.48       |
> |     Sunglasses    |     User-Trained-DM    |      1004.71 ± 51.23    |     8.50 ± 0.45    |     1.52 ± 0.03    |        43.48 ± 5.70       |         57.96 ± 1.60       |
> |     Sunglasses   |       Fine-tuning      |      747.69 ± 59.18     |     4.15 ± 0.52    |     1.40 ± 0.02    |        71.64 ± 4.66       |         71.12 ± 2.85       |
> |     Sunglasses    |  CG    |     1195.85 ± 121.26    |     8.07 ± 1.80    |     1.71 ± 0.15    |        73.82 ± 23.67      |        27.24 ± 12.20       |
> |     Sunglasses    |   Domain-Guidance |      710.53 ± 30.45     |     4.06 ± 0.16    |     1.43 ± 0.06    |        87.38 ± 15.07      |         75.44 ± 2.71       |
> |     Sunglasses    |        MMD (Ours)      |      692.87 ± 30.43     |     3.25 ± 0.18    |     1.39 ± 0.04    |        113.13 ± 9.36      |         79.08 ± 1.54       |
>
>
> |    User    |     Guidance |     FD ↓    |         KD ↓       |  RRKE ↓ |      Density (×100) ↑     |      Coverage (×100) ↑     |
> |:---------:|:-------:|:-----------:|:---------:|:-------:|:--------:|:------:|
> |     Reading-glasses    |      No-Guidance-DM    |   702.83 ± 27.10     |     2.22 ± 0.16    |     1.35 ± 0.01    |        68.79 ± 13.60      |         80.96 ± 2.50       |
> |     Reading-glasses    |     User-Trained-DM    |  1105.12 ± 53.03     |     7.54 ± 0.29    |     1.47 ± 0.01    |        59.14 ± 4.18       |         57.52 ± 2.91       |
> |     Reading-glasses    |   Fine-tuning  |       732.91 ± 43.02     |     2.99 ± 0.20    |     1.35 ± 0.01    |        82.82 ± 4.47       |         73.40 ± 1.89       |
> |     Reading-glasses    |    CG   |       678.48 ± 21.16     |     2.16 ± 0.14    |     1.34 ± 0.01    |        70.12 ± 12.81      |         77.96 ± 1.28       |
> |     Reading-glasses    |    Domain-Guidance     |       667.56 ± 21.04     |     2.08 ± 0.13    |     1.34 ± 0.01    |        75.74 ± 14.97      |         81.72 ± 2.30       |
> |     Reading-glasses    |   MMD (Ours)   |       574.29 ± 17.57     |     1.39 ± 0.05    |     1.30 ± 0.01    |        87.10 ± 9.69       |         84.60 ± 2.54       |
>
>
> **2- W2: MMD Guidance for adapting highly different domains**
>
> We appreciate the reviewer’s point on the training-free nature of MMD guidance and whether this can hinder the application of MMD guidance for adapting to very different domains. First, we would like to clarify that the prompt-aware experiments in the work, which are visualized in the Appendix (currently in Figures 17,18,19), represent adaptation tasks between relatively distant domains.
>
> To further test MMD guidance in highly different domains, following Reference [1], we used two distinct painting styles, Klimt and Franz Marc, as the reference set and applied MMD guidance to adapt the distribution of samples generated by SDXL. To quantitatively measure the domain gap between SDXL outputs and the reference set, we computed Recall and Coverage, obtaining values of 0.34 and 0.12, respectively. In the following Table, we report results with/without MMD guidance.
>
>
> | Model  | FD $\downarrow$ | KD $\downarrow$  | RRKE $\downarrow$ | Density (×10²) $\uparrow$ | Coverage (×10²) $\uparrow$ |
> |:---:|:-----:|:---------:|:-------:|:--------:|:----:|
> | SDXL  (No Guidance)  | 1988.48 | 4.38  | 1.98   | 9.83    | 12.76      |
> | SDXL + MMD (Ours)  | 1564.35 | 2.28  | 1.74   | 24.34   | 52.74   |
>
> ---
>
> [1] Somepalli et al., “Measuring Style Similarity in Diffusion Models”. ECCV 2024

---

> ### Author Response · Authors · 2025-11-26
> **Official Comment by Authors (continued)**
>
> **3- W3: Effect of reference sample size on the MMD Guidance Results**
>
> Following the question of the reviewer, we have numerically tested the performance of MMD guidance and report the results in the following table for the experiments of Figures 2, 3 in the submission. As can be seen, the performance changes significantly when the number of reference samples varies from 0 (no MMD guidance) to 100 and then the performance looks mildly improving with more reference data. We will include the table in the revised draft.
>
> **Table 1:** The following are the numerical results for the FFHQ case:
>
> |User|#Reference Samples|FD↓|KD↓|RRKE↓|
> |:----:|:------:|:----:|:----:|:----:|
> |Sunglasses|0 (No MMD Guidance)|1004.71|8.21|1.73|
> |Sunglasses|10|886.41|5.67|1.55|
> |Sunglasses|50|736.69|3.78|1.44|
> |Sunglasses|150|719.37|3.59|1.41|
> |Sunglasses|250|710.48|3.33|1.41|
> |Sunglasses|400|696.07|3.29|1.38|
> |Sunglasses|500|692.87|3.25|1.39|
>
> **Table 2:** Here are the numerical results for the prompt-aware SD-XL case:
>
> | # of reference samples | Guidance | FD $\downarrow$ | KD $\downarrow$ | RRKE $\downarrow$ |
> |:-----:|:---:|:----:|:------:|:------:|
> | n = 0         | No Guidance | 1953.75  |    3.57         | 1.93              |
> | n = 10        |    MMD      | 1805.92  | 2.7657          | 1.89              |
> | n = 50        |    MMD      | 1731.27 | 2.6882          |  1.83             |
> | n = 100       |    MMD      | 1691.42   | 2.5682          |  1.81             |
> | n = 150       |    MMD      | 1686.61  | 2.5301          |  1.80             |
> | n = 200       |    MMD      | 1674.45         | 2.4945          |  1.79             |
>
> **Table 4:** The following are the numerical results for the synthetic case of Gaussian mixtures:
>
> | | | |8 Gaussian Component|||25 Gaussian Component||
> |-|-|:-:|:-:|:-:|:-:|:-:|:-:|
> |**#References**|**Guidance**|**FD $\downarrow$** |**KD(1000) $\downarrow$**| **RRKE $\downarrow$** | **FD $\downarrow$**| **KD(1000) $\downarrow$** | **RRKE $\downarrow$** |
> |0|No Guidance|7.04|142.42|2.31|77.70|330.91|2.073|
> |5|MMD|4.62|65.17|1.45|29.63|79.72|0.81|
> |25|MMD|0.49|29.72|1.33|4.93|22.66|0.33|
> |50|MMD|0.38|24.51|1.30|3.52|9.49|0.23|
> |100|MMD|0.34|23.33|1.22|3.69|11.25|0.25|
> |150|MMD|0.37|17.02|1.29|3.56|9.52|0.23|
>
>
> **4- Effect of Noisy reference samples**
>
> Since we typically consider a limited size of reference samples to apply MMD guidance for distributional adaptation, we rely on the correctness of estimating the reference data distribution from the limited number of reference samples. As a result, MMD guidance could potentially perform less satisfactorily under significant **adversarial** distribution shifts.
>
> On the other hand, considering random (non-adversarial) additive noise for contaminating the reference data, our numerical results show that MMD guidance would be relatively robust to moderate levels of additive noise. To support this, we have run an experiment to test this and report the performance of MMD guidance applied to noisy data of the paper’s experiments of SD-XL data in the following table with different Gaussian noise $\mathcal{N}(\mathbf{0},\sigma^2 I)$ parameter $\sigma$ values.
>
> The table below reports FD, KD, and RRKE for different noise levels, showing that these metrics degrade only slightly as the noise level increases from 0% to 30%, which indicates that MMD guidance maintains its effectiveness under moderate random corruption of the reference set.
>
> | % of noise   | Guidance | FD $\downarrow$ | KD $\downarrow$ | RRKE $\downarrow$ |
> |:---:|:---:|:---:|:------:|:---:|
> | 0 (No noise) | MMD      | 1674.45         | 2.4945          | 1.79              |
> | 10%          | MMD      | 1681.72         | 2.5042          | 1.82              |
> | 20%          | MMD      | 1688.70         | 2.5162          | 1.85              |
> | 30%          | MMD      | 1698.52         | 2.5237          | 1.87              |
> | *0 (No noise)* | *No Guidance* | *1953.75*         | *3.57*          | *1.93*         |
>
> **5- MMD Guidance and robustness to gradient hacking**
>
> To answer the reviewer's question, we would like to highlight a key property of MMD distance which can make it a suitable choice for the domain adaptation guidance. MMD distance essentially measures the Hilbert-norm difference of the expected values of a representation of data in the corresponding Hilbert space of the kernels. The reason that this property matters is that the MMD loss summarizes the data estimated from the data to the expected value of the corresponding representation in the Hilbert space, i.e., the MMD loss is invariant to the changes of the distribution orthogonal to the mean vector in the corresponding Hilbert space. We will include this discussion in the revision.

---

### Official Review · Reviewer_eNzo · 2025-11-05

**Soundness:** 2
**Presentation:** 2
**Contribution:** 2
**Rating:** 2
**Confidence:** 2

**Summary:**

The paper studies the mismatch between generated samples from diffusion models and target data. To address this issue, the authors augment the generative process with gradients of the maximum mean discrepancy (MMD) between generated samples and a reference dataset. The MMD-based method is practical since it doesn't suffer the curse of dimensionality. Furthermore, this method extends to prompt-aware distribution matching and latent diffusion models. Experiments are provided to verify the distribution matching performance.

**Strengths:**

- The authors use the maximum mean discrepancy (MMD) between generated samples and a reference dataset to address the distribution mismatching problem. This is a natural approach to guide generation toward our desired distribution.

- The MMD is a practical metric that measures the discrepancy between generation and reference distributions, since it avoids the curse of dimensionality from other metrics.

- The authors provide an explicit evaluation of MMD gradient and illustrate the necessity of cross term for encouraging distribution matching.

- The MMD-based method extends to prompt-aware distribution matching, which is a widely used application: text-image generation. Experiments are also provided to demonstrate the distribution matching performance.

**Weaknesses:**

- The maximum mean discrepancy is used to measure the distribution mismatching at every step. This might not be the best way. The reason is that  initially distribution mismatching should be large, and gradually reduces as generation ends. We only need to ensure the final  few steps have small distribution mismatching.

- As shown in Theorems 1 and 2, it takes a large number of reference data to ensure that the empirical cross term is close to the ideal one. Since this is required for all time steps, it might require a large reference data set.

- The MMD-based method is not analyzed, theoretically. It is not characterized how MMD-based guidance push the generative process to a target distribution.

- Experiments show the distribution matching performance is attained by sacrificing image background quality.

**Questions:**

See comments in Weaknesses.

---

> ### Author Response · Authors · 2025-11-26
>
> We sincerely thank Reviewer eNzo for the thoughtful feedback and questions on our work. Below, we provide our responses and clarification regarding the comments raised. We will be happy to further discuss the points during the discussion phase.
>
> **1- W1: Application of MMD guidance at every step**
>
> We would like to clarify that we do not necessarily require the application of MMD guidance at every iteration of the reverse diffusion process. Similar to other existing guidance schemes, the coefficient of the MMD guidance can be managed with a scheduling process. We will clarify this point in the revision.
>
> **2- W2, Part 1: Theoretical statements and the sample complexity for estimating the empirical cross term**
>
> We would like to clarify that Theorem 1 *does not imply* a large reference sample requirement. Indeed, both the theorems indicate that the MMD gradient can be estimated accurately with a *moderate* number of reference samples. This is made explicit in Corollary 1 for the RBF (Gaussian) kernel, which provides a *dimension-free* bound of order
>
> $$\mathcal{O}\left(\frac{1}{\sigma\sqrt{N_r}}\right),$$
>
> where $\sigma$ is the kernel bandwidth and $N_r$ is the number of reference samples. Importantly, this bound controls the **$L_2$-norm of the full gradient vector**, yet remains independent of the ambient dimension. Therefore, we clarify that neither the theorem statement nor the corollary support the interpretation that the MMD guidance method requires an excessively large reference dataset.
>
> **3- W2, Part 2: “Since this is required for all time steps, it might require a large reference data set.”**
>
> We think there may be a misunderstanding regarding the role of the number of time steps $T$ in the concentration bounds. Note that the bounds in Theorem 1 (Eq. 8) and Corollary 1 (Eq. 9) hold with probability $1-\delta$ and depend on $\delta$ only through the term $\sqrt{\log(1/\delta)}$.
>
> When applying the bound across $T$ steps, we only need each step to hold with probability $1 - \frac{\delta}{T}$. By a standard union bound argument, the bound then holds **simultaneously** for all $T$ steps with probability $1-\delta$. Consequently, the factor $\sqrt{\log(1/\delta)}$ becomes
>
> $$
> \sqrt{\log\left(\frac{T}{\delta}\right)},
> $$
>
> introducing only a mild multiplicative $\sqrt{\log T}$ factor. To illustrate this scale, for a large-scale case of $T = 10^4$ steps for generating $M = 10^4$ images, the extra multiplicative factor is only $\sqrt{\log(MT)} < 5$, which would be minor in practice. Therefore, the number of time steps *does not lead to* a large reference dataset requirement. We will include a remark in the revision to clarify this logarithmic dependence.

---

> ### Author Response · Authors · 2025-11-26
> **Official Comment by Authors (continued)**
>
> **4- W3: Theoretical analysis of MMD guidance**
>
> We would like to clarify that Theorems 1–3 and Corollaries 1–2 prove that the empirical MMD guidance direction concentrates around the true gradient direction of the population MMD distance $\text{MMD}(P_{\text{ref}}, P_{\text{gen}})$. Since MMD is an established discrepancy measure for probability distributions, these results provide a theoretical justification for using its estimated gradient as a guidance signal in diffusion sampling. In particular, Theorem 1 and Corollary 1 guarantee that the estimated guidance direction remains accurate under moderate sample sizes and does not deteriorate with dimensionality.
>
> In addition to the theoretical results, our experiments demonstrate that MMD guidance consistently improves or maintains performance across multiple state-of-the-art latent diffusion models, in both prompt-free and text-guided settings. We believe that the combination of these concentration guarantees and empirical outcomes supports the soundness and utility of the proposed MMD guidance method.
>
> **5- W4: Image background quality in MMD guidance**
>
> We would like to respectfully request the reviewer to refer us to the specific figure they believe shows “sacrificing image background quality.” After re-checking the figures, we do not observe a considerable loss in background quality that is specifically caused by MMD guidance. This is also supported by the quantitative results, as both FD and KD scores of MMD guidance are consistently better than those of the baselines.
>
> We would also like to highlight the diversity of the base latent diffusion models (LDMs) in our experiments, whose unguided visual qualities are already different. Our experiments include both the early, smaller-scale LDM architecture of Rombach et al. [1]  and the more recent, large-scale text-to-image models SD-XL [2] and PixArt-$\Sigma$ [3]. These models have different native visual qualities even before applying MMD guidance. Therefore, a fair evaluation of the quality of generated images should take the original model’s baseline performance into account.
>
> ---
>
> [1] Rombach et al. “High-Resolution Image Synthesis With Latent Diffusion Models”. CVPR 2022
>
> [2] Podell et al. “SDXL: Improving Latent Diffusion Models for High-Resolution Image Synthesis”. ICLR 2024
>
> [3] Chen et al. “PixArt-$\Sigma$: Weak-to-Strong Training of Diffusion Transformer for 4K Text-to-Image Generation”. ECCV 2024

---

### Author Response · Authors · 2025-12-03
**Wrap-up Summary of Author Responses and Discussion-Phase Updates**

We sincerely thank the reviewers for their constructive feedback and questions. We have uploaded a revised manuscript, with the updates highlighted in blue, to improve clarity and presentation in line with the points addressed in our responses to the reviewers. Below, we summarize the key clarifications and updates provided during the discussion phase.

**1. Theoretical Concentration Guarantees in the Manuscript**

We clarified that our theoretical results in Theorems 1,3 and Corollaries 1,3 provide **dimension-free concentration bounds** showing that the empirical MMD guidance direction accurately estimates the population MMD gradient. As highlighted in our response and discussed in Remark 2, the application of these bounds across $T$ diffusion steps **introduces only a mild $\mathcal{O}(\sqrt{\log T})$ factor**, indicating that MMD guidance *does not require large reference sets* and remains theoretically stable over long sampling trajectories.

**2. Empirical Behavior of MMD Guidance across Reference Sample Sizes**

In response to the reviewers, we provided several ablations in FFHQ, SDXL, and synthetic Gaussian mixture settings. Our numerical evaluations show that the performance of MMD Guidance improves sharply from *0 to 50–100 reference samples* and then stabilizes, supporting that **moderate reference sizes can be sufficient in practical applications**, which also aligns with our theoretical sample-complexity guarantees.

**3. Robustness to Noisy and Complex Reference Distributions**

We numerically demonstrated that MMD guidance could perform robustly against *moderate-level random noise* in the reference samples, and remains effective under *considerable domain shifts* (e.g., distinct painting styles). We also clarified that we have already conducted MMD-Guidance with *complex GPT-4o-produced prompts*. These results indicate that MMD guidance can maintain stable performance across a broad range of realistic, noisy, and distributionally distant adaptation scenarios.

**4. Comparison to Domain Guidance and Training-Free Baselines**

Following Reviewer 5JBa’s suggestion, we added the *Domain-Guidance* baseline and found that while it performs better than several existing baselines, *MMD guidance could yield a better result, particularly in correcting mode-weight mismatches*. We also discussed recent training-free methods (β-CFG, CFG++, learnable scheduling) and clarified that our approach is *fully training-free*, offering a complementary and general-purpose distribution-matching mechanism for standard and latent diffusion models.

**5. Mixture-Component Proportion Correction via MMD Guidance**

Following Reviewer zL3j’s suggestion, we numerically demonstrated through Gaussian mixture experiments that *MMD guidance can accurately correct mixture-component proportions*, aligning the generated samples with the reference mixture weights. This highlights a key advantage of MMD guidance over training-based or fine-tuning approaches, as it can directly adjust distributional mass without retraining the underlying diffusion model.

---

### Meta-Review · Area_Chair_Ft6D · 2026-01-06

**Summary:**

This paper initially received three negative reviews and one positive review. In response, the authors provided clarifications and additional experiments that address many of the reviewers’ concerns.

However, while several specific weaknesses appear to have been reasonably resolved, it is less clear whether these improvements are sufficient to change the reviewers’ initial negative assessments. Beyond addressing individual issues, it is also important to consider the paper’s overall quality and its potential impact on the community. From this broader perspective, I am not fully convinced that the paper meets the acceptance threshold for ICLR this year. That said, the authors’ efforts are evident, and I encourage the authors to further refine and strengthen the draft for a future submission.

**Reviewer Concerns:**

The following summarizes a few concerns raised by the reviewers.

[Reviewer eNzo]
Most of the concerns were addressed, with the exception of the necessity of enforcing MMD matching at every training step. While the authors provided a reasonable justification, it is not entirely clear whether this fully resolves the issue. The reviewer may still expect a more rigorous or analytical treatment to better substantiate this design choice.

[Reviewer 5JBa]
The missing reference was acknowledged and added during the rebuttal. However, the omission appears to have left a negative initial impression. In addition, the reviewer questioned the effectiveness of the method under large domain gaps, and may expect evaluations on more clearly differentiated or challenging domain pairs.

[Reviewer 7KWN]
Concerns regarding additional hyperparameters, more complex prompts, and potential extensions to other modalities were addressed through supplementary experiments and examples provided in the rebuttal.

**Reviewer Scores:**

Please see the sections above.

---

### Decision · Program_Chairs · 2026-01-26

Reject